# Aerosol retrievals from different polarimeters during the ACEPOL campaign using a common retrieval algorithm

Guangliang Fu[1], Otto Hasekamp[1], Jeroen Rietjens[1], Martijn Smit[1], Antonio Di Noia[2], Brian Cairns[3], Andrzej Wasilewski[4], David Diner[5], Felix Seidel[5], Feng Xu[6], Kirk Knobelspiesse[7], Meng Gao[7], Arlindo da Silva[7], Sharon Burton[8], Chris Hostetler[8], John Hair[8], and Richard Ferrare[8]

[1]Netherlands Institute for Space Research (SRON, NWO-I), Utrecht, The Netherlands.
[2]University of Leicester.
[3]NASA Goddard Institute for Space Studies (GISS).
[4]Trinnovim LLC.
[5]NASA Jet Propulsion Laboratory (JPL).
[6]School of Meteorology, The University of Oklahoma National Weather Center.
[7]NASA Goddard Space Flight Center.
[8]NASA Langley Research Center.

**Correspondence:** Guangliang Fu (g.fu@sron.nl), Otto Hasekamp (o.hasekamp@sron.nl)

**Abstract.**

In this paper, we present aerosol retrieval results from the ACEPOL (Aerosol Characterization from Polarimeter and Lidar) campaign, which was a joint initiative between NASA and SRON - Netherlands Institute for Space Research. The campaign took place in October-November 2017 over the western part of the United States. During ACEPOL six different instruments

were deployed on the NASA ER-2 high altitude aircraft, including four Multi-Angle Polarimeters (MAPs): SPEX airborne, the Airborne Hyper Angular Rainbow Polarimeter (AirHARP), the Airborne Multi-angle SpectroPolarimeter Imager (AirMSPI), and the Research Scanning Polarimeter (RSP). Also, two lidars participated: the High Spectral Resolution Lidar -2 (HSRL-2) and the Cloud Physics Lidar (CPL). Flights were conducted mainly for scenes with low aerosol load over land but also some cases with higher AOD were observed. We perform aerosol retrievals from SPEX airborne, RSP (410-865 nm range only), and

AirMSPI using the SRON aerosol retrieval algorithm and compare the results against AERONET and HSRL-2 measurements (for SPEX airborne and RSP). All three MAPs compare well against AERONET for the Aerosol Optical Depth (AOD) (Mean Absolute Error (MAE) between 0.014-0.024 at 440 nm). For the fine mode effective radius the MAE ranges between 0.021-0.028 micron. For the comparison with HSRL-2 we focus on a day with low AOD (0.02-0.14 at 532 nm) over the California Central Valley, Arizona and Nevada (26 October) and a flight with high AOD (including measurements with AOD > 1.0 at

532 nm) over a prescribed forest fire in Arizona (9 November). For the day with low AOD the MAE in AOD (at 532 nm) with HSRL-2 are 0.014 and 0.022 for SPEX and RSP, respectively, showing the capability of MAPs to provide accurate AOD retrievals for the challenging case of low AOD over land. For the retrievals over the smoke plume also a reasonable agreement in AOD between the MAPs and HSRL-2 was found (MAE 0.088 and 0.079 for SPEX and RSP, respectively), despite the fact that the comparison is hampered by large spatial variability in AOD throughout the smoke plume. Also a good comparison is

found between the MAPs and HSRL-2 for the aerosol depolarization ratio (a measure for particles sphericity) with MAE of

0.023 and 0.016 for SPEX and RSP, respectively. Finally, SPEX and RSP agree very well for the retrieved microphysical and optical properties of the smoke plume.

## 1  Introduction

Aerosols such as smoke, sulphate, dust, and volcanic ash particles affect the Earth climate directly by interaction with radiation and indirectly by modifying the cloud properties. In contrast to the warming effect of greenhouse gases, which is understood quite well, the quantification of aerosol cooling contains a large uncertainty, as reported in the latest (5th) assessment report of the Intergovernmental Panel on Climate Change (IPCC, 2014). This large uncertainty adds substantial difficulties in the prediction of the Earth's climate change in future. Aerosols also have a big influence on air quality. Air pollution from aerosols may result in severe adverse problems to human health (Wyzga and Rohr, 2015). To improve our understanding of the aerosol effect on climate and air quality, accurate global measurements of aerosol optical properties (e.g., aerosol optical depth (AOD), single scattering albedo (SSA)), microphysical properties (size distribution, refractive index, particles shape), and their vertical distribution, are of crucial importance. Satellite instruments are needed to obtain such measurements at a global scale.

Lidar measurements are needed to obtain vertical profile information about aerosols. The Cloud-Aerosol Lidar with Orthogonal Polarization (CALIOP) elastic backscatter Lidar (Winker et al., 2010), has been providing aerosol Lidar measurements since 2006. High Spectral Resolution Lidar (HSRL) techniques (Hair et al., 2008) are being used for the new generation of Lidar instrumentation such as the Cloud-Aerosol Transport System (CATS) instrument (Yorks et al., 2014), which has been operational on the International Space Station (ISS) in the period 2015-2017, and for the European Space Agency (ESA) Earthcare mission (Illingworth et al., 2014), expected for launch in 2021. In comparison to elastic backscatter lidars, the HSRL technique has an additional filtered channel that provides an assessment of aerosol extinction. It also improves the accuracy of the aerosol backscatter profile, especially at altitudes far from the instrument, since it is calculated as a direct ratio of two channels instead of retrieved with assumptions that result in accumulating errors. The HSRL methodology also improves the aerosol depolarization through the improved backscatter and provides the aerosol lidar ratio using the extinction (Burton et al., 2012).

From a passive remote sensing point-of-view, instruments that measure both intensity and polarization and observe a ground pixel under multiple viewing angles contain the richest set of information about aerosols in our atmosphere (Dubovik et al., 2019). The reason is that the angular dependence of the scattering matrix elements related to linear polarization, depend strongly on the microphysical aerosol properties, like refractive index and particle size (Hansen and Travis, 1974; Mishchenko and Travis, 1997). Furthermore, the polarization signal is mostly dominated by light that has been scattered only once, which means that the characteristics of the scattering matrix remain largely preserved in a top-of-atmosphere polarization measurement. The added value of polarization has been demonstrated by a number of studies on synthetic measurements (Mishchenko and Travis, 1997; Hasekamp and Landgraf, 2007; Hasekamp, 2010; Knobelspiesse et al., 2012), airborne measurements (Chowdhary et al., 2005; Waquet et al., 2009; Xu et al., 2017; Wu et al., 2015, 2016), and spaceborne measurements (Hasekamp et al., 2011b; Dubovik et al., 2011; Fu and Hasekamp, 2018). These algorithms can be divided in two main groups: LookUp-Table (LUT)

based approaches and full inversion approaches. Generally speaking, LUT approaches are faster but less accurate than full inversion approaches because LUT approaches choose the best fitting aerosol model from a discrete lookup table. Full inversion approaches are more accurate but slower because they require radiative transfer calculations as part of the retrieval procedure. The LUT algorithms are e.g., the LOA LUT algorithm over ocean (Deuzé et al., 2000), the LOA LUT algorithm over land (Deuzé et al., 2001; Herman et al., 1997), and the SSA LUT algorithm (Waquet et al., 2016). The full inversion algorithms are e.g., the GRASP algorithm (Dubovik et al., 2011), the SRON-Aerosol algorithm (Hasekamp and Landgraf, 2007; Hasekamp et al., 2011b; Stap et al., 2015; Wu et al., 2015, 2016; Di Noia et al., 2017; Fu and Hasekamp, 2018), the JPL algorithm (Xu et al., 2017), the GISS algorithm (Waquet et al., 2009) and the MAPP algorithm (Stamnes et al., 2018). Besides, some additional aerosol retrieval approaches can be found in (Sano et al., 2006; Cheng et al., 2011; Masuda et al., 2000; Lebsock et al., 2007). It should be noted that of the full inversion approaches only the SRON-Aerosol algorithm and the GRASP algorithm have been applied at a global scale.

The best known satellite instruments that performed multi-angle photopolarimetric measurements of the Earth atmosphere were the POLDER (Polarization and Directionality of the Earth's Reflectances) instruments (Deschamps et al., 1994), of which the recently decommissioned POLDER-3 on board the PARASOL micro-satellite provided data from 2005-2013. Although the original algorithms for aerosol retrieval from POLDER-3 do not make full use of the information contained in the MAP measurements (Deuzé et al., 2000, 2001), algorithms developed more recently (Dubovik et al., 2011; Hasekamp et al., 2011b; Fu and Hasekamp, 2018) do fully exploit the available information and provide insight in the capabilities and limitations of the POLDER-3 instrument. The advanced data products of these algorithms have been applied at global (Lacagnina et al., 2015, 2017) and regional (Chen et al., 2018) scale. The main limitation of the POLDER instruments is the limited accuracy with which the Degree of Linear Polarization (DoLP) can be measured. The DoLP accuracy is intrinsically limited by the filter wheel technology, which relies on sequential measurements of different polarization directions, in combination with a spatial under-sampling. On the other hand, the advantage of this technology is that it allows for a large swath with (almost) global coverage in a day. The POLDER design also forms the blueprint for the 3MI instruments (Fougnie et al., 2018), to be flown on METOP-SG in the time frame ~2020-2035.

Focus for the development of new polarimetric instrumentation has been on improved polarimetric accuracy, more viewing angles, more wavelengths, an extended spectral range, or a combination of these aspects. For a number of these instrument concepts airborne demonstrators for possible future satellite missions have been built: 1) the Research Scanning Polarimeter (RSP) (Cairns et al., 2004) which is an airborne version of the Aerosol Polarimetry Sensor (APS) (Mishchenko et al., 2007) that was lost in a failed launch in 2011. RSP measures at many viewing angles (~150) and 9 wavelength bands between 410-2250 nm. It has a demonstrated DoLP accuracy of better than 0.002 (Knobelspiesse et al., 2019). 2) The Airborne Multiangle SpectroPolarimetric Imager (AirMSPI) (Diner et al., 2013). AirMSPI is an eight-band (355, 380, 445, 470, 555, 660, 865, 935 nm) pushbroom camera, measuring polarization in the 470, 660, and 865 nm bands, mounted on a gimbal to acquire multiangular observations over a $\pm 67°$ along-track range. The AirMSPI concept will be implemented in a satellite mission as the Multi-Angle Imager for Aerosols (MAIA) to be launched in ~2021 (Diner et al., 2018). 3) The Airborne Hyper-Angular Rainbow Polarimeter (AirHARP) (Martins et al., 2018). AirHARP is a wide field-of-view imager that measures in

the spectral bands at 440, 550, 670, and 865 nm where 670 nm is measured under 60 and the other bands under 20 viewing geometries. This concept will be implemented in a satellite instrument for a Cubesat mission to be launched in 2019 and for the Phytoplankton Aerosol Cloud and ocean Ecosystems (PACE) mission, to be launched 2022 (Werdell et al., 2019). 4) The Spectro-polarimeter for Planetary Exploration (SPEX airborne) instrument (Smit et al., 2019). SPEX airborne employs the
5 spectral modulation technique (Snik et al., 2009) to accurately measure the DoLP with a spectral resolution of 10-20 nm. The intensity is being measured at higher spectral resolution of 2-3 nm. SPEX airborne performs multi-angle measurements at 9 viewing angles ranging between $\pm 56°$ in a spectral range between 400-800 nm. The SPEX concept will be implemented in a satellite instrument SPEXone (Hasekamp et al., 2019) for the NASA PACE mission (Werdell et al., 2019).

All 4 airborne MAPs listed above were mounted on the NASA Earth Resources-2 (ER-2) high altitude ($\sim 20$ km) aircraft
(Navarro, 2007) during the Aerosol Characterization from Polarimeter and Lidar (ACEPOL) campaign, which was performed from October-November 2017, starting from the NASA Armstrong airbase in Palmdale, California. During ACEPOL, also two lidars were deployed on the ER-2: the High Spectral Resolution Lidar-2 (HSRL-2) (Hair et al., 2008), providing vertically resolved measurements of backscatter coefficients (at 355, 532, and 1064 nm), extinction coefficients (at 355 and 532 nm), and depolarization ratio (at 355, 532, and 1064 nm) and the Cloud Physics Lidar (CPL) (McGill et al., 2002), providing vertically
resolved measurements of backscatter coefficients at 355, 532, and 1064 nm and depolarization ratio at 1064 nm.

The goals of the ACEPOL campaign include: (i) comparison of level-1 (radiance and DoLP) performance between the different MAPs, (ii) comparison of aerosol retrievals from the different MAPs, (iii) comparing MAP retrievals to lidar retrievals, and (iv) performing combined retrievals using both MAP and lidar measurements. The focus of this paper is on aspects (ii) and (iii): We will perform aerosol retrievals from RSP, SPEX airborne, and AirMSPI measurements during ACEPOL, and
20 evaluate the retrievals against AERONET and against HSRL-2. Note that aerosol retrievals from AirHARP measurements are not included in this paper, because the data were not available when performing the here presented analysis.

In this study, we evaluate the performance of the different MAPs for retrieving aerosol optical and microphysical properties, and also their capabilities to provide lidar related aerosol properties. The retrieved aerosol properties are validated and compared with the data from AERONET and HSRL-2. The paper is organized as follows. Section 2 introduces the methodologies
of the SRON algorithm for polarimetric aerosol retrievals, section 3 describes the data sets from the ACEPOL campaign, which are used in this study, and the retrievals of different MAPs from ACEPOL are performed and compared with AERONET and HSRL-2 in section 4. Finally, the last section summarizes and concludes this study.

## 2 Methodology

### 2.1 SRON multimode retrieval algorithm

In this paper, we employ the SRON aerosol retrieval algorithm in multimode setup (Fu and Hasekamp, 2018). In principle, the idea of the multimode approach is that instead of fitting the size distribution parameters (the effective radius $r_{\text{eff}}$ and the effective variance $v_{\text{eff}}$) of two modes, one aims to fit the size distribution with a larger number of modes for which $r_{\text{eff}}$ and $v_{\text{eff}}$ are fixed. The advantage of this approach is that it makes the inversion problem more linear since $r_{\text{eff}}$ and $v_{\text{eff}}$ tend to

make the inversion highly nonlinear. Another advantage is that the multimode approach has more freedom in fitting different shapes of size distribution if the number of chosen modes is sufficiently large. In this paper, multimode retrievals based on 5 modes are used and the aerosol size distribution are described in Table 1 (Fu and Hasekamp, 2018). We consider mode 1-3 together as the fine mode and mode 4 and 5 together as the coarse mode. To account for spectral dependence, we describe the

5 refractive index $m$ for the fine and coarse mode as $m(\lambda) = \sum_{k=1}^{n_\alpha} \alpha_k\, m^k(\lambda)$ where $m^k(\lambda)$ are prescribed refractive indices as function of wavelength and $\alpha_k$ are coefficients to be determined in the retrieval (see below). Both real part and imaginary part of refractive indices are represented in this way. Here, we base the spectral dependence of the refractive index of the standard types of D'Almeida et al. (1991) (Inorganic/Sulphate, Black Carbon, and Dust). The coefficients $\alpha_k$ are the real numbers between 0 and 1, and are defined as weighting factors to combine the refractive index spectra for different aerosol

components, e.g., DUST, water (H2O), Black Carbon (BC), INORGanic matter (INORG). In this study, we set $n_\alpha = 2$ and assume that spectral dependence of the fine mode and the coarse mode refractive indices can be described respectively by INORG+BC and DUST+INORG. Note that this assumption is flexible and can be updated according to the information content of the measurement. Also spectra based on Principal Component Analysis (PCA) can be used as in Wu et al. (2015). The standard refractive index spectra are only used to describe the spectral dependence as the MAP measurements do not contain

sufficient information to retrieve the refractive index for each wavelength separately. Nonspherical aerosols are described as a size/shape mixture of randomly oriented spheroids (Hill et al., 1984; Mishchenko et al., 1997). We use the Mie/T-matrix-improved geometrical optics database by Dubovik et al. (2006) along with their proposed spheroid aspect ratio distribution for computing optical properties for a mixture of spheroids and spheres. The aerosol parameters included in the retrieval state vector $x$ are the aerosol column numbers for the 5 modes (Table 1), 2 coefficients (Inorganic, Black Carbon) for the fine

mode refractive index, 2 coefficients (Inorganic, Dust) for the coarse mode refractive index, the fraction of spherical particles (assumed the same for all modes), and the central height of a Gaussian aerosol height distribution (assumed the same for all modes).

For the surface reflection matrix we use (Rahman et al., 1993; Litvinov et al., 2011; Xu et al., 2017):

$$\mathbf{R}_s(\lambda, \mu_{\text{in}}, \mu_{\text{out}}, \phi_v - \phi_0) = A(\lambda) \left( \frac{(\mu_{\text{in}}\, \mu_{\text{out}})^{k-1}}{(\mu_{\text{in}} + \mu_{\text{out}})^{1-k}}\, F(g, \Theta)[1 + R(G)] \right) \mathbf{D} + \mathbf{R}_{\text{pol}} \tag{1}$$

$$\mathbf{R}_{\text{pol}}(\mu_{\text{in}}, \mu_{\text{out}}, \phi_v - \phi_0) = B \left( \frac{\exp\left(-\tan(\frac{\pi - \Theta}{2})\right)\, \exp\left(-\nu\right)\, F_p(m, \Theta)}{4(\mu_{\text{in}} + \mu_{\text{out}})} \right) \tag{2}$$

where $k$ is a parameter that varies between 0 and 1. This parameter controls the slope of the reflectance with respect to the illumination and view angles (Rahman et al., 1993). $\mathbf{D}$ is the null matrix except $\mathbf{D}_{11} = 1$. The first part in Eq. (1) accounts for the bidirectional reflectance distribution function (BRDF) parameterized by the Rahman-Pinty-Verstraete (RPV) model

(Rahman et al., 1993). The pairs $(\theta_0, \phi_0)$ and $(\theta_v, \phi_v)$ respectively denote the solar and viewing zenith and azimuth angles. $\mu_{\text{in}}$ and $\mu_{\text{out}}$ are respectively the cosines of incoming and outgoing angles. $g$ is the asymmetry parameter of the Henyey-Greenstein phase function $F(g, \Theta)$. $\Theta$ is the scattering angle. $1 + R(G)$ is an approximation of the hot spot effect (Rahman et al., 1993), where $G = \sqrt{\tan^2\theta_0 + \tan^2\theta_v - 2\tan\theta_0 \tan|\theta_v| \cos(\phi_v - \phi_0)}$ and $R(G) = \frac{1 - A(\lambda)}{1 + G}$. The second part in Eq. (1) accounts for

the surface polarized reflectance, where we use the model proposed by Maignan et al. (2009). $\mathbf{R}_{\text{pol}}$ is expressed by Eq. (2) (as stated by Eq. (31) in Litvinov et al. (2011)). Here, $B$ is a scaling parameter (band-independent). $F_p(m, \Theta)$ is the element $F_{21}$ of the Fresnel scattering matrix with refactive index $m$. Parameter $\nu$ is taken based on Atmospherically Resistant Vegetation Index (ARVI) (Kaufman and Tanre, 1992). Here we use $\nu = 0.6$. Based on Eq. (1) and Eq. (2), We include $A(\lambda)$ at each measured wavelength, and $k$, $g$, and $B$ as fit parameters in the state vector. In this paper, we perform aerosol retrievals from SPEX airborne, RSP, and AirMSPI, and the state vectors corresponding to these three polarimeters are listed in Table 2.

The measurement vector $\boldsymbol{y}$ contains the measured radiances (sun normalized) and Degree of Linear Polarization (DoLP) values at the different wavelengths and viewing angles. To retrieve the state vector from the measurements, a damped Gauss-Newton iteration method with Phillips-Tikhonov regularization is employed (Hasekamp et al., 2011b; Fu and Hasekamp, 2018). The inversion algorithm finds the solution $\hat{\boldsymbol{x}}$, which solves the minimization-optimization problem,

$$\hat{\boldsymbol{x}} = \min_{\boldsymbol{x}}(||\mathbf{S}_y^{-\frac{1}{2}}(\mathbf{F}(\boldsymbol{x}) - \boldsymbol{y})||^2 + \gamma^2||\mathbf{W}^{-\frac{1}{2}}(\boldsymbol{x} - \boldsymbol{x}_{\text{a}})||^2). \tag{3}$$

Here, $\mathbf{F}$ is the forward model that simulates the measurement for a given state vector $\boldsymbol{x}$. $\mathbf{F}$ consists of a radiative transfer model, for which we use the SRON radiative transfer model LINTRAN Landgraf et al. (2001); Hasekamp and Landgraf (2002, 2005); Schepers et al. (2014). All the radiative transfer calculations are performed for a model atmosphere that includes Rayleigh scattering, scattering and absorption by aerosols, and gas absorption. Rayleigh scterring cross sections are used from Bucholtz (1995). The forward model simulates Stokes parameters $I, Q, U$ at the height of the observation (e.g., $\sim 20$ km for NASA ER-2 in this paper) for given optical properties (scattering and absorption optical thickness and scattering phase matrix for each vertical layer of the model atmosphere ( - 15 layers of atmosphere is assumed ). The other part of the forward model computes the optical properties from the aerosol microphysical properties using the tabulated kernels of Dubovik et al. (2006) for a mixture of spheroids and spheres.

Since the forward model is nonlinear the inversion problem has to be solved iteratively replacing the forward model in each iteration step by its linear approximation,

$$\mathbf{F}(\boldsymbol{x}) \approx \mathbf{F}(\boldsymbol{x}_n) + \mathbf{K}(\boldsymbol{x} - \boldsymbol{x}_n). \tag{4}$$

Here, $\mathbf{K}$ is the Jacobian matrix (with $K_{ij} = \frac{\partial F_i}{\partial x_j}(\boldsymbol{x}_n)$), which contains the derivatives of the forward model with respect to each variable in the state vector $\boldsymbol{x}$. Therefore, the optimization problem (Eq. (3)) is reduced to

$$\tilde{\boldsymbol{x}}_{n+1} = \min_{\tilde{\boldsymbol{x}}}(||\tilde{\mathbf{K}}(\tilde{\boldsymbol{x}} - \tilde{\boldsymbol{x}}_n) - \tilde{\boldsymbol{y}}||^2 + \gamma^2||\tilde{\boldsymbol{x}} - \tilde{\boldsymbol{x}}_a||^2), \tag{5}$$

where $\tilde{\mathbf{K}} = \mathbf{S}_y^{-\frac{1}{2}}\mathbf{K}\mathbf{W}^{\frac{1}{2}}$, $\tilde{\boldsymbol{x}} = \mathbf{W}^{-\frac{1}{2}}\boldsymbol{x}$ and $\tilde{\boldsymbol{y}} = \mathbf{S}_y^{-\frac{1}{2}}(\boldsymbol{y} - \mathbf{F}(\boldsymbol{x}_n))$. $\boldsymbol{x}_{\text{a}}$ is the a priori state vector, $\mathbf{W}$ is a weighting matrix that ensures that all state vector parameters range within the same order of magnitude (Hasekamp et al., 2011b), and $\mathbf{S}_y$ is the measurement error covariance matrix. Table 2 shows the values of $\boldsymbol{x}_{\text{a}}$ for aerosol and surface parameters. $\mathbf{W}$ is a diagonal matrix and its diagonal values are also shown in Table 2 (in the "weight" column). The solution of Eq. (5) is given by:

$$\tilde{\boldsymbol{x}}_{n+1} = \tilde{\boldsymbol{x}}_n + \Lambda(\tilde{\mathbf{K}}^T\tilde{\mathbf{K}} + \gamma^2\mathbf{I})^{-1}(\tilde{\mathbf{K}}^T\tilde{\boldsymbol{y}} - \gamma^2(\tilde{\boldsymbol{x}}_n - \tilde{\boldsymbol{x}}_a)). \tag{6}$$

$\Lambda$ is a filter/damping factor, which limits the step size for each iteration of the state vector. In this way, we use a Gauss-Newton scheme with reduced step size to avoid diverging retrievals (Hasekamp et al., 2011a). The filter factor $\Lambda$ values between 0 and 1. The regularization parameter $\gamma^2$ in Eq. (3) is chosen optimally (for each iteration) from different values (5 values from 0.1 to 5) by evaluating the goodness of fit using a simplified (fast) forward model. In the SRON aerosol algorithm, the first guess is obtained before the full inversion retrieval using a multimode Look-Up Table (LUT), which is based on tabulated RT calculations for each mode. The pre-calculated LUT is used as input for an approximate forward model in the LUT retrieval. Here, single scattering is computed exactly as its computational cost is negligible. The fit parameters in the LUT retrieval are the aerosol column numbers for each mode and the surface parameters. For the first guess of the refractive index we use a fixed value of 1.45 for all modes. For further details we refer to Fu and Hasekamp (2018).

We use the goodness of fit ($\chi^2$) to decide whether the retrievals have successfully converged:

$$\chi^2 = \frac{1}{n_{\text{meas}}} \sum_{i=1}^{n_{\text{meas}}} \frac{(F_i - y_i)^2}{S_y(i,i)}. \tag{7}$$

Here, $n_{\text{meas}}$ is the total number of measurements (multi-angle and multispectral radiance and DoLP) for each pixel. We consider valid retrievals those that achieve a $\chi^2$ smaller than an empirically chosen threshold $\chi^2_{\text{max}}$. This filter rejects cases in which the forward model is not able to fit the measurements, e.g., because of cloud-contaminated pixels (Stap et al., 2015, 2016), corrupted measurements (Hasekamp et al., 2011b), and cases in which the first guess state vector deviates too much from the truth (Di Noia et al., 2015).

## 2.2 Fine mode and coarse mode effective radius

According to Eq. (2.53) in (Hansen and Travis, 1974), the effective radius is defined:

$$r_{\text{eff}} = \frac{\int_{r_{\text{min}}}^{r_{\text{max}}} \pi r^3 n(r) dr}{\int_{r_{\text{min}}}^{r_{\text{max}}} \pi r^2 n(r) dr} = \frac{R}{O} \tag{8}$$

where $n(r)dr$ is the number of particles with radius between $r$ and $r + dr$. $r_{\text{min}}$ and $r_{\text{max}}$ are the particle radius for the smallest and largest particles.

In this study, a 5-mode retrieval is used. The effective radius for multiple modes together ($r_{\text{eff}}^{\text{m}}$) is calculated from the different fixed modes by: $r_{\text{eff}}^{\text{m}} = \dfrac{\sum_{n^{\text{m}}} R_i}{\sum_{n^{\text{m}}} O_i}$ where $n^{\text{m}}$ is the number of modes grouped together. For the 5-mode retrievals in this study, we compute $r_{\text{eff}}$ for the fine mode (modes 1-3 together) and and coarse mode (modes 4 and 5 together).

## 2.3 Aerosol depolarization ratio and aerosol lidar ratio

The aerosol lidar properties are related to the aerosol scattering matrix. For some general assumptions ((i) scattering by an assembly of randomly oriented particles each having a plane of symmetry, (ii) scattering by an assembly containing particles and their mirror particles in equal numbers and with random orientations, (iii) Rayleigh scattering with or without depolarization effects), the aerosol scattering matrix has a simplified block-diagnonal structure (Bottiger et al., 1980; Mishchenko, 2014):

$$
\mathbf{F}(\Theta) = \begin{bmatrix} F_{11}(\Theta) & F_{12}(\Theta) & 0 & 0 \\ F_{12}(\Theta) & F_{22}(\Theta) & 0 & 0 \\ 0 & 0 & F_{33}(\Theta) & F_{34}(\Theta) \\ 0 & 0 & -F_{34}(\Theta) & F_{44}(\Theta) \end{bmatrix}
\tag{9}
$$

where $\Theta$ is the scattering angle and $F_{11}$ is the phase function for total radiance.

The aerosol (linear) depolarization ratio is defined as:

$$
\delta_{\mathrm{col}}^{\mathrm{pol}} = \frac{F_{11}(180°) - F_{22}(180°)}{F_{11}(180°) + F_{22}(180°)}
\tag{10}
$$

which is adpated from Eq. (3) in (Mishchenko et al., 2016). We use Eq. (10) to compute an aerosol depolarization ratio from the aerosol properties of the MAPs and compare this to the vertically integrated value measured by HSRL-2, which is calculated by:

$$
\hat{\delta}^{\mathrm{hsrl}}(i) = \frac{\delta^{\mathrm{hsrl}}(i)}{1 + \delta^{\mathrm{hsrl}}(i)}, \qquad \hat{\delta}_{\mathrm{col}}^{\mathrm{hsrl}} = \frac{\sum\limits_{i=0}^{n_{\mathrm{bin}}}(\hat{\delta}^{\mathrm{hsrl}}(i)\,\beta_{\mathrm{b}}^{\mathrm{hsrl}}(i))}{\sum\limits_{i=0}^{n_{\mathrm{bin}}}(\beta_{\mathrm{b}}^{\mathrm{hsrl}}(i))}, \qquad \delta_{\mathrm{col}}^{\mathrm{hsrl}} = \frac{\hat{\delta}_{\mathrm{col}}^{\mathrm{hsrl}}}{1 - \hat{\delta}_{\mathrm{col}}^{\mathrm{hsrl}}},
\tag{11}
$$

where $i = 0$ corresponds to the bin closet to the surface, $i = n_{\mathrm{bin}}$ corresponds to the bin closet to the aircraft. The aerosol
backscatter coefficient ($\beta_{\mathrm{b}}^{\mathrm{hsrl}}(i)$) for each bin is used as the weighting parameter. $\delta^{\mathrm{hsrl}}(i)$ is first transformed to $\hat{\delta}^{\mathrm{hsrl}}(i)$, which is because $\hat{\delta}^{\mathrm{hsrl}}(i)$ mix linearly like backscatter, but $\delta^{\mathrm{hsrl}}(i)$ does not (Burton et al., 2014).

In our retrieval algorithm we assume that for aerosols the single scattering albedo $\omega$ and $F_{11}$ do not depend on altitude. In that case, using $\omega$ and $F_{11}(180°)$, we compute the vertically integrated aerosol extinction-to-backscatter ratio, i.e., aerosol lidar ratio for a MAP by:

$$
S_{\mathrm{col}}^{\mathrm{pol}} = \frac{4\pi}{\omega\,F_{11}(180°)},
\tag{12}
$$

which is adpated from Eq. (4) in Lopes et al. (2013). This can be compared to the corresponding value from HSRL-2:

$$
S_{\mathrm{col}}^{\mathrm{hsrl}} = \frac{\sum\limits_{i=0}^{n_{\mathrm{bin}}}(\beta_{\mathrm{e}}^{\mathrm{hsrl}}(i))}{\sum\limits_{i=0}^{n_{\mathrm{bin}}}(\beta_{\mathrm{b}}^{\mathrm{hsrl}}(i))}
\tag{13}
$$

which is adapted from Eq. (28) in Stamnes et al. (2018). Here $\beta_{\mathrm{e}}^{\mathrm{hsrl}}(i)$ denotes the extinction coefficient for each bin.

## 3  Measurements

For this study, we use airborne measurements from 3 different polarimeters (SPEX airborne, RSP, AirMSPI) and one lidar (HSRL-2). Further, we use ground based measurements for validation and re-analysis data as input to our retrieval algorithm. All data are described in this section.

## 3.1 RSP

RSP (Cairns et al., 1999) started to operate on the NASA ER-2 since 2010 and has flown on a number of other airplanes since 2001 (Cairns et al., 2003). Multi-viewing capability over a large along-track angular range and at many viewing angles ($\sim 150$) is obtained using a scanning mirror. Due to the fact that some viewing angles are blocked by the aircraft, the angular range of RSP on the ER-2 is restricted to -40° to 60°. The Stokes parameters $Q$ and $U$ are analyzed in separate refractive telescopes, using Wollaston prisms, followed by dichroic beamsplitters. The RSP instrument is equipped with an in-flight calibration system, and the accuracy for the DoLP is better than 0.002 (Knobelspiesse et al., 2019), providing a benchmark (in DoLP) for other MAPs. Aerosol retrievals from RSP have been performed, amongst others, by Waquet et al. (2009); Wu et al. (2015, 2016); Di Noia et al. (2017); Stamnes et al. (2018); Gao et al. (2019).

A complicating factor for using RSP measurements in aerosol retrievals over inhomogeneous land surfaces is that different viewing angles have different ground pixel size and may look at slightly shifted scenes on the ground. To partly overcome this problem we use (1) the approach of Wu et al. (2015) and construct RSP pixels that represent a 5 km along track running average; (2) the (moving average) approach of Di Noia et al. (2017) to select 10 viewing angles covering a broad viewing angle range (over the total RSP viewing angles) and convolve RSP measurements at each selected angle with an average of 5 angles. In this sense, although averaged measurements 10 viewing angles are input to the retrieval algorithm, they are constructed from original RSP measurements at 50 angles. In this study, we use 5 wavelengths (410, 469.1, 554.9, 670, and 863.4 nm) for RSP retrievals as Di Noia et al. (2017). The viewing angles and wavelengths used in retrievals are summarized in Table 2. It should be noted that theoretically the SWIR bands of RSP 1590 and 2250 nm would provide extra constraints for the characterization of coarse mode aerosols. For the ACEPOL campaign however, we found no improvement by including the SWIR bands, and even slightly worse results (compared to AERONET and HSRL-2) in some cases. A possible explanation is that our assumption that the directional property of surface reflection is spectrally neutral does not hold over the full RSP wavelength range. Another explanation may be that the SWIR channels are affected by gas absorption which we could not perfectly correct for.

## 3.2 AirMSPI

AirMSPI (Diner et al., 2013) started to operate on the NASA ER-2 since October 2010. AirMSPI is an eight-band (355, 380, 445, 470, 555, 660, 865, 935 nm) pushbroom camera, which measures linear polarization in the 470, 660, and 865 nm bands. AirMSPI employs a photoelastic modulator-based polarimetric imaging technique to enable accurate measurements of Degree of Linear Polarization (DoLP) in addition to intensity. The instrument is mounted on a gimbal to acquire multiangular observations in the range of $\pm 67°$. AirMSPI has two principal observing modes: (1) step-and-stare, where 11 km$\times$ 11 km targets are observed at a discrete set of view angles with a spatial resolution of $\sim 10$ m. (2) continuous sweep, where the camera slews back and forth along the flight track between $\pm 65°$ to acquire wide area coverage (11 km swath at nadir, target length 108 km). The spatial resolution is $\sim 25$ m. Aerosol retrievals from AirMSPI have been performed by Xu et al. (2017, 2018, 2019). In this study, only the step-and-stare measurements have been used as they provide a mult-angle-view of the same ground scene. For ACEPOL, AirMSPI was programmed to measure at 9 viewing angles in the step-and-stare mode: 0° (nadir),

$\pm 29°$, $\pm 48°$, $\pm 59°$, $\pm 66°$. Radiance measurements are used at all wavelengths except 935 nm and DoLP measurements at all 3 wavelengths. The viewing angles and wavelengths used in retrievals are summarized in Table 2. Following Xu et al. (2017), for AirMSPI we aggregate individual ground pixels to 1 km × 1 km spatial grid in order to be less affected by surface inhomogeneity and its effect on the angular co-registration.

## 3.3 SPEX airborne

SPEX airborne performed its first (engineering) flight on the ER-2 in 2016. ACEPOL has been the first full science campaign. The instrument employs the spectral modulation technique (Snik et al., 2009) to accurately measure the Degree of Linear Polarization (DoLP) in the spectral range 400-800 nm with a spectral resolution of 10-20 nm, and the intensity at a higher spectral resolution of 2-3 nm. A ground-based version of SPEX has performed upward looking measurements from the ground which have been used to successfully retrieve aerosol microphysical and optical properties by van Harten et al. (2014); Di Noia et al. (2015). SPEX airborne performs multi-angle measurements at 9 viewing angles: $\pm 56°$, $\pm 42°$, $\pm 28°$, $\pm 14°$, and $0°$. Smit et al. (2019) performed a comparison between SPEX airborne and RSP for radiance and DoLP measurements at 410, 470, 550, and 670 nm. They found very good agreement between SPEX airborne and RSP at 550 and 670 nm whereas the agreement gets worse towards smaller wavelengths. In this study, we use measurements of radiance and DoLP at 16 wavelengths, (450, 460, 470, 480, 490, 500, 510, 520, 530, 540, 550, 565, 580, 600, 670, and 750 nm). The measurement at each wavelength represents an average of a 10 nm wide spectral region. We leave out the shortest wavelengths because of less good agreement with RSP, and the wavelengths >750 nm because of order overlap of the grating. The viewing angles and wavelengths used in retrievals are summarized in Table 2.

Each SPEX viewport has a moderate swath of $\sim 6$ degrees (Smit et al., 2019) in the across-track direction, which translates to a projected field of view from 2.4 km at nadir to 4.5 km at fore and aft viewports when the instrument is operated at the typical altitude of ER-2. Conceptually, the instrument acts as nine separate pushbroom spectrometers, which produce nine overlapping strips of data on the ground. In this way, a multi-angular view is obtained of ground scenes when the aircraft flies over it. The spatial sampling of the L1C product is chosen as 1 km × 1 km (across × along-track), which is driven by the L1B spatial resolution of the outer viewports.

## 3.4 HSRL-2 data

The NASA Langley HSRL-2 instrument, operational since 2012, is a successor to the NASA Langley airborne HSRL-1 instrument, which was described by Hair et al. (2008); Burton et al. (2012) and validated by Rogers et al. (2009). The HSRL-2 uses the HSRL technique to independently measure aerosol extinction and backscatter at 355 nm (Burton et al., 2018) and 532 nm and the standard backscatter technique to measure aerosol backscatter at 1064 nm (Müller et al., 2014). It is polarization sensitive at all three wavelengths. HSRL-2 measures vertically resolved values for the backscatter coefficient ($\beta$) and aerosol depolarization ratio at 355, 532, and 1064 nm (Burton et al., 2015) and the extinction coefficient and AOD at the high-spectral-resolution channels, 355 and 532 nm. HSRL-2 is the first airborne system capable of providing 3 backscatter and 2 extinction measurements, which is important for lidar retrievals of microphysical properties (Müller et al., 2014).

For the ACEPOL flights on the ER-2, the aerosol backscatter coefficient is derived using the HSRL technique at 355 nm and 532 nm and the elastic backscatter technique at 1064 nm and reported at a vertical resolution of 15 m and a horizontal/temporal resolution of 10 seconds (approximately 1-2 km at ER-2 cruise speeds). The aerosol depolarization ratios at all 3 wavelengths are reported at the same resolutions. For ACEPOL, the extinction products from the HSRL method are reported at 150 m vertical

resolution and at temporal resolution of 60 s generally and 10 s. Additionally, the aerosol extinction products at 355 nm and 532 nm are also provided based on the aerosol backscatter and an assumed lidar ratio of 40 sr, and reported at the backscatter resolution.

Similarly, the AOD is reported from the standard HSRL approach and also the AOD calculated using the assumed lidar ratio is provided. The reason why two AOD products are reported is that during ACEPOL, HSRL-2 experienced an interference that

appears to be related to atmospheric turbulence. This interference impacted the ability to use the 532 nm and 355 nm molecular channels to derive aerosol extinction and AOD from the usual HSRL method. However, this interference did not impact the measurements of aerosol backscatter profiles and so these profiles were computed using the HSRL technique (i.e. ratio of total backscatter to molecular backscatter). The systematic uncertainties on the AOD from the HSRL method is about 0.05 for ACEPOL, whereas the assumed lidar ratio produces systematic uncertainty that is a constant relative fraction ($\pm 50\%$).

Therefore, for the case with high AOD (i.e., Figs 7 and 8), the uncertainty is smaller when using the HSRL method and we therefore use these products for this case. Conversely, although the uncertainties are fairly high for both products for low AOD, the product using an assumed lidar ratio is expected to have lower uncertainties, and we use these products for the low AOD cases (i.e., Figs 4, 5, and 6) in this paper.

### 3.5 AERONET data

The multispectral aerosol optical depth (AOD) from the MAP and lidar retrievals is validated with AERONET (AErosol RObotic NETwork) level 1.5 data (Holben et al., 2001) (version 3.0). The data are cloud cleared. The uncertainty on AERONET AOD is 0.01 for mid-visible wavelengths and 0.03 for UV wavelengths (Eck et al., 1999) and is dominated by a calibration (systematic) error. The effective radius for fine and coarse modes are compared with AERONET level 1.5 Almucantar Retrieval Inversion Products (Dubovik et al., 2002). The AOD of fine and coarse modes are compared with AERONET level 1.5 spectral

de-convolution algorithm (SDA) data (O'Neill et al., 2003). It should be noted that the inversion- and SDA products are quite uncertain themselves at low AOD so the comparison to these products should not be considered a validation. In this paper, data from the 6 following AERONET stations are used for validation: Bakersfield, CalTech, Flagstaff ("USGS_Flagstaff_ROLO"), Fresno_2, Modesto, and Railroad-Valley.

### 3.6 Re-analysis data

The required meteorological inputs for our retrieval scheme are vertical profiles of humidity, temperature, and pressure. We obtain these information from National Centers for Environmental Prediction (NCEP) reanalysis data (Kalnay et al., 1996). For ozone absorption in retrievals, we use the ozone profiles from Modern-Era Retrospective analysis for Research and Ap-

plications, Version 2 (MERRA-2) (Gelaro et al., 2017). The $NO_2$ columns are taken from Air Force Geophysics Laboratory (AFGL) database. The data are interpolated to the specific time and location of a MAP ground pixel.

## 4   Results

We apply the SRON algorithm as described in section 2.1 to measurements of SPEX, RSP, and AirMSPI. In our retrievals we use an ad-hoc representation of the measurement error covariance matrix $\mathbf{S}_y$, where we assume a diagonal matrix for $\mathbf{S}_y$ (i.e. errors are uncorrelated for different wavelengths and viewing angles) with values on the diagonal corresponding to 5 % error on the radiance and 0.005 on DoLP. Although this is a crude assumption that does not reflect a bottom-up estimate taking into account individual error sources, it should be noted that for the chosen inversion approach the most important aspect is the relative dependence between radiance and DoLP errors, because we include a flexible regularization parameter that is determined as part of the retrieval. The same results can be obtained when assuming 2.5 % radiance and 0.0025 DoLP errors, in combination with a different $\chi^2$ filter. This relative dependence between radiance and DoLP accuracies seems reasonable for all three instruments given that they all have a high DoLP accuracy. Another note is that there are error sources such as mis-registration between different viewing angles, that are not included in an uncertainty model (as they are not directly related to pure instrument performance) but that are significant and possibly even dominant over land.

To compare MAP retrievals with AERONET or HSRL-2, $\chi^2 < 1.5$ is used in this paper (for SPEX, RSP, and AirMSPI) as the filter for the goodness of fit. Besides, we also apply filters on the number of viewing angles ($\geq 9$), the smallest scattering angle ($< 120°$), and the largest scattering angle ($> 120°$). To evaluate the retrieved aerosol properties, three measures are used, which are the Mean Absolute Error (MAE), the Mean Relative Error (MRE), the bias, and the STandard Deviation (STD). Two types of plots are included in this paper for comparisons. One is the scatter plot with x- and y-axis respectively for two instruments. The other one is the Bland-Altman (Martin Bland and Altman, 1986) plot (difference plot), where the differences between two instruments are plotted against the the averages of the two intruments.

### 4.1   SPEX airborne, RSP, and AirMSPI versus AERONET

We first compare the polarimetric (SPEX, RSP, and AirMSPI) retrievals with the AERONET data for the aerosol optical depth (AOD) at three wavelengths 380 nm, 440 nm, and 675 nm. For the comparison, retrievals within 10 km around each AERONET station are selected and averaged. The AERONET data are averaged within 1 hour around the time of the ER-2 overpass. The results of the AOD comparison are shown in Figure 1 where panels a,d correspond to SPEX airborne, panels b,e to RSP, and panels c,f to AirMSPI. The 12 overpasses between SPEX and AERONET are consistent with those between RSP and AERONET, i.e., one averaged value from SPEX (Fig 1a) and RSP (Fig 1b) correspond to the same averaged value from AERONET. For AirMSPI (Fig 1c), 8 overpasses are consistent with SPEX and RSP, while the other 4 comparison points do not have corresponding points for SPEX and RSP. The reason for this inconsistency in comparison points is that AirMSPI was not making measurements for some of the AERONET overpasses. On the other hand some of the SPEX and RSP overpasses

are screened out because there were no ground pixels with enough co-located viewing angles because of aircraft yaw, while the swath of AirMSPI is sufficiently large to still get co-located angles despite the yaw.

For the AOD at 440 nm, the MAE is respectively 0.016, 0.024, 0.014, the MRE is respectively 0.175, 0.289, 0.139, the bias is respectively 0.003, -0.010, -0.004, and the STD is respectively 0.019, 0.027, 0.017 for SPEX, RSP, and AirMSPI. The MAE, bias, and STD are within 0.01 and the MRE is within 0.15 for the instruments, where the values for SPEX airborne and AirMSPI are somewhat smaller than for RSP. Similar conclusions hold for the AOD at 380 nm or 675 nm. For each instrument, the MAE gets smaller with increasing wavelengths, which is mainly caused by the fact that the AOD value itself decreases with wavelength. Based on the comparisons above, we can conclude that the SPEX, RSP, and AirMSPI all achieve good agreement with AERONET and the differences in performance between the instruments are small.

For the comparison of the fine and coarse mode effective radius ($r_{\mathrm{eff}}^{\mathrm{f}}$ and $r_{\mathrm{eff}}^{\mathrm{c}}$), it should be noted that it is difficult to retrieve them when AOD is small. Therefore, shown in Figure 2 are the comparison when $\tau_{380}$ is larger than 0.1. The remaining cases are still very challenging but we would lose too many points if we further increase the AOD limit. The solid lines shown in the plot are bias±STD. The retrievals of $r_{\mathrm{eff}}^{\mathrm{f}}$ compared with AERONET are shown in Figs. 2a-c, where the MAE is 0.022, 0.021, and 0.028 $\mu$m for SPEX, RSP, and AirMSPI, respectively. SPEX and RSP compare somewhat better in terms of MAE and bias whereas AirMSPI has a small STD. However, overall the differences between the instruments are small and the number of comparison points is very limited, which means that differences between instruments can be explained by 1 or 2 points. The $r_{\mathrm{eff}}^{\mathrm{c}}$ comparisons corresponding to SPEX, RSP, and AirMSPI are shown in Figs 2d-f, respectively. All three instruments have a poor comparison with AERONET for $r_{\mathrm{eff}}^{\mathrm{c}}$, with a MAE close to 1.5 $\mu$m. This is in line with synthetic studies (e.g., Hasekamp et al. (2019)) that $r_{\mathrm{eff}}^{\mathrm{c}}$ is a difficult parameter to retrieve, in particular for small AOD values. It should be noted that AERONET consistently gives larger coarse mode effective radius than MAPs. A possible explanation is that the effective radius for the coarse mode 4 and 5 in our 5-mode retrieval are 0.882 and 1.719 respectively (see Table 1), thus the coarse mode effective radius from MAPs calculated based on Eq. (8) is estimated and limited between 0.882 and 1.719, wheras AERONET gives values between 2.25 and 3.3 (when $\tau_{380}$ is larger than 0.1). A comparable range is expected for MAPs if a parametric 2-mode retrieval or a $\geq$ 7-mode retrieval (Fu and Hasekamp, 2018) is used. Also, it should be noted that the "fine" and "coarse" as defined by the Almucantar retrievals are different with defining "fine" and "coarse" by specific modes as shown in Table 1. This may introduce differences in the comparisons.

For the comparison of the fine and coarse mode AOD ($\tau^{\mathrm{f}}$ and $\tau^{\mathrm{c}}$), the results are shown in Figure 3. The comparison shows a MAE of 0.028, 0.029, and 0.012 for SPEX, RSP, and airMSPI, respectively for $\tau^{\mathrm{f}}$ and 0.026, 0.028, 0.017 for $\tau^{\mathrm{c}}$. The bias is 0.028, 0.019 and 0.004 for $\tau^{\mathrm{f}}$ and 0.025, 0.028, and 0.003 for $\tau^{\mathrm{c}}$. So, SPEX and RSP have an overestimation of the fine mode and an underestimation of the coarse mode, compared to AERONET SDA product. Although these biases are large in a relative sense (given the low AOD, especially for the coarse mode), they are within the expected error from the AERONET SDA product. AirMSPI compares better to the AERONET SDA product than SPEX airborne and RSP. Again, it should be noted that the AirMSPI comparison does not contain exactly the same points as the comparison for SPEX and RSP. It should also be noted that for RSP, since no SWIR channels are included in the retrieval to nail the coarse mode, we thus don't expect it to do well retrieving coarse mode properties. It is important to note that for the low AOD values encountered during ACEPOL, the

AERONET retrieved fine and coarse mode AOD and effective radius are very uncertain themselves. Therefore, this comparison should not be interpreted as "retrieval versus truth" but rather as "retrieval versus retrieval".

## 4.2 Comparison between SPEX airborne, RSP, and HSRL-2

For the comparison to HSRL-2, we only use SPEX airborne and RSP because these provided a continuous data stream during ACEPOL, while AirMSPI only provides step-and-stare measurements for specific targets.

### 4.2.1 Comparison HSRL-2 to AERONET

Given that we use HSRL-2 as a reference for our MAP retrievals, it is important to first validate HSRL-2 with AERONET. Figure 4 shows the comparison of the HSRL-2 AOD at 355 nm and 532 nm with AERONET (log-log interpolated between 340 and 380 nm for 355 nm and between 500 and 675 nm for 532 nm). From the comparison it follows that the HSRL-2 AOD at 532 nm agrees very well with AERONET, with a small MAE (0.012), a small MRE (0.269) a small absolute bias (0.005), and a small STD (0.014). The comparison at 355 nm is somewhat worse than that at 532 nm with an MAE of 0.028, an MRE of 0.357, a bias of -0.014, and a STD of 0.029. The bias between HSRL-2 and AERONET is within the AERONET uncertainty. The random differences, with standard deviation 0.029 at 380 nm and 0.014 at 532 nm are most likely due to HSRL-2 uncertainties. Note that shown in Figure 4 are the points corresponding to days 23 October, 25 October, 26 October, and 7 November 2017.

### 4.2.2 Low AOD case on 26 October 2017

In this subsection, we compare the aerosol properties from SPEX and RSP with those from HSRL-2 for the day 26 October 2017 with low aerosol loading (AOD at 532 nm in the range 0.02 to 0.14). The results for AOD at 355 nm and 532 nm are shown in Figure 5. Figure 5a shows the retrieved AOD from HSRL-2 for the ground pixels co-located with SPEX and RSP. From this figure it follows that there were very low AOD values for the eastern part of the scene and somewhat higher values in the western and south-western part of the scene. Figure 5b shows the AOD comparison between SPEX and HSRL-2 with the MAE 0.014, the MRE 0.296, the bias 0.009, and the STD 0.018 at 532 nm, and the MAE 0.028, the MRE 0.321, the bias -0.006, and the STD 0.034 at 355 nm. Figure 5c shows the AOD comparison between RSP and HSRL-2 with the MAE 0.022, the MRE 0.418, the bias -0.007, and the STD 0.028 at 532 nm, and the MAE 0.037, the MRE 0.369, the bias -0.008, and the STD 0.048 at 355 nm. So, SPEX shows a very good agreement with HSRL-2 for this challenging scene of low AOD over land with a relatively bright surface. SPEX compares somewhat better to HSRL-2 than RSP for this case at both 532 and 355 nm. The Bland-Altman plots Figs 5e and f show a larger scatter and more outliers for RSP. A possible explanation is that for low AOD the radiance and polarization measurements have strong influence from the spatially inhomogeneous surface, and therefore errors due to inter-angle mis-registration, which are larger for RSP than for SPEX, may be significant. For these cases there is larger sensitivity to spatial mismatch between different viewing angles, and RSP, as a single-pixel-swath instrument, is more sensitive to such mismatches. Figure 5d shows the AOD comparison between SPEX and RSP with the MAE 0.024, the MRE 0.831, the bias 0.016, and the STD 0.025. The differences from the direct comparison between SPEX and RSP are

somewhat larger than those from individual comparisons with HSRL-2 of SPEX and RSP, respectively. This suggests that the differences with HSRL-2 are not caused by common assumptions in the SPEX and RSP retrievals, but are rather caused by errors that are specific to each MAP.

For the retrieved surface parameters, we do not have a good reference to evaluate the accuracy. Instead, Figure 6 shows the
AOD difference between MAP and HSRL-2 as function of retrieved BRDF scaling parameter $A$, where we do not see clear correlation or dependence.

### 4.2.3   High AOD on 9 November 2017

In this subsection, polarimetric retrievals from SPEX airborne and RSP are compared to HSRL-2 on the day 9 November 2017 for a smoke plume with high AOD (including AOD values > 1.0). Figure 7a shows the original AOD (i.e., no filter or colocation
included) from SPEX for the flight leg over the smoke plume. This gives a sense of how variable the smoke plume is. Figure 7b shows the AOD comparison between SPEX and HSRL-2, where the MAE is 0.088, the MRE is 0.693, the bias is -0.029, and the STD is 0.149 at 532 nm. Figure 7c shows the AOD comparison between RSP and HSRL-2, where the MAE is 0.079, the MRE is 0.564, the bias is -0.024, and the STD is 0.142 at 532 nm. Figure 7d shows the AOD comparison between SPEX and RSP, where the MAE is 0.044, the MRE is 0.155, the bias is -0.005, and the STD is 0.063 at 532 nm. RSP compares slightly
better to HSRL-2 than SPEX with respect to MAE and MRE. It should be noted that the smoke plume exhibits large spatial variation so part of the MAP-lidar differences can be attributed to the fact that different instruments see a slightly different part of the smoke plume. Furthermore, both SPEX and RSP show a similar negative bias in AOD at both 355 nm and 532 nm, and one clear outlier point in the comparison with HSRL-2 at the highest AOD. This is also clear from the corresponding Bland-Altman plots Figs 7e and f. Given the very similar underestimation in both SPEX and RSP (compared to HSRL-2) and the
good comparison between SPEX and RSP, it is unlikely that this underestimation is caused by aspects related to instrumental errors of the 2 different MAPs. It might be possible that the underestimation is related to the MAP retrieval approach which is the same for both instruments, but based on earlier studies with real and synthetic measurements we have no indication for this. Another possibility is that HSRL-2 overestimates the AOD at this high aerosol loading or the large spatial variability has a larger effect on the MAP-lidar comparison than on the inter-MAP comparison. At high AOD the performance of RSP is more
similar to that of SPEX than for low AOD. Our explanation for this, is that at high AOD the measured radiance and DoLP are less affected by the co-registration errors between viewing angles than for low AOD.

For the high AOD case, we compare also the aerosol depolarization ratio ($\delta$) and aerosol lidar ratio ($S$) from SPEX and RSP with HSRL-2. Figs 8a-c respectively show the comparison of the aerosol depolarization ratio between SPEX and HSRL-2, between RSP and HSRL-2, and between SPEX and RSP. It can be observed that both SPEX and RSP show a similar behavior
against HSRL-2 especially at 355 nm: There is an underestimation towards lower values of depolarization ratio but on the other hand there is a reasonable agreement with HSRL-2 for both instruments. Again, given the fact that the performance of both SPEX and RSP versus HSRL-2 is very similar, we conclude that the main reason for difference between SPEX/RSP and HSRL-2 does not lie in instrumental errors for the MAPs. A possible explanation for the difference could be the simplified description of non-spherical particles in our retrieval approach. On the other hand, the overall comparison of the aerosol depolarization

ratio with HSRL-2 confirms capability of both SPEX and RSP to retrieve information on particle shape. The results of the aerosol lidar ratio are shown in Figs 8d-f. Both SPEX and RSP show a similar overestimation of the lidar ratio compared to HSRL-2, and SPEX and RSP agree quite well. Again, it is unlikely that the overestimation is related to instrumental errors in the MAPs. Overall, the agreement with HSRL-2 for the aerosol lidar ratio is reasonable for both SPEX and RSP.

### 4.2.4    Median and standard deviation properties of the smoke plume

The median and standard deviation properties for the smoke plume as measured by SPEX and RSP are summarized in Table 3. Here, we only include retrievals for which AOD > 0.2 at 532 nm because for those cases accurate retrieval of microphysical properties is expected (e.g., Hasekamp et al. (2019)). The number of points to calculate the median and the standard deviation is the same for SPEX and RSP. Also, we only include fine mode microphysical properties because there is only a very small coarse mode contribution to the smoke plume. SPEX and RSP compare well for the fine mode refractive index, fine mode effective redius, fine and coarse mode AOD, and SSA (relative to requirements as formulated e.g. by Mishchenko et al. (2004)). Reasonable agreement is found for the fraction of spherical particles. For the Aerosol Layer Height (ALH), SPEX retrieves a higher value (4.417 km) than RSP (1.148 km), where the latter value is somewhat closer to the ALH derived from HSRL-2 (2.64 km). Here, it should be noted that for SPEX the shortest wavelength that is used in the retrieval is 450 nm, so we do not expect an accurate ALH retrieval because the retrieval of ALH from polarization requires a strong signal from Rayleigh scattering (Wu et al., 2016). Figure 9 shows the number particle size distribution from SPEX and RSP in the smoke plume, which confirms the smoke plume is fine mode dominated.

The values of the aerosol properties in Table 3 (for both SPEX and RSP) are in the range that is expected for smoke. First of all, it is expected that smoke is dominated by fine particles (e.g. Russell et al. (2014)), which is confirmed by the much larger retrieved fine mode AOD than coarse mode AOD by both MAPs. The real part of the refractive index is consistent with the study of Levin et al. (2010) for the Fire Laboratory at Missoula Experiment (FLAME), who found mostly refractive indices for biomass burning between 1.55 and 1.60. Also, the SSA values in Table 3 are representative for fresh biomass burning smoke. For example, Nicolae et al. (2013) found SSA values of 0.79 at 532 nm for smoke with an age of 0.25 day and 0.93 for smokes with an age of 0.75 day. Both the values retrieved by SPEX and RSP can be considered realistic for smoke.

## 5    Discussions and conclusions

In this study, we performed aerosol retrievals from different MAPs employed during the ACEPOL campaign and evaluated them against ground based AERONET measurements and against HSRL-2 measurements. The polarimetric aerosol retrievals were performed using the SRON algorithm in multi-mode setup (Fu and Hasekamp, 2018) on SPEX airborne, RSP (without SWIR channels), and AirMSPI.

For the AERONET comparison, only scenes with low AOD (0.03-0.17 at 440 nm) were available during ACEPOL. For these scenes, SPEX, RSP, and AirMSPI all show good agreement with AERONET for AOD (MAE respectively 0.016, 0.024, and 0.014 for AOD at 440 nm). For the fine mode effective radius, we found MAE with AERONET of 0.022, 0.021, and 0.028

for SPEX, RSP, and AirMSPI, respectively. For the effective radius comparison we only compare scenes with AOD > 0.10 at 380 nm, but it should be noted that the remaining cases are still very challenging and that the difference in performance between the different instruments are caused by just 1 or 2 comparison points. All three instruments had a poor comparison with AERONET for the coarse mode effective radius. This was because the coarse mode effective radius was a difficult parameter to retrieve, in particular for small AOD values. For the fine mode AOD, good agreements with AERONET were shown for all three MAPs with somewhat better performance for AirMSPI. For the coarse mode AOD, SPEX and RSP show reasonable agreement while AirMSPI shows also good agreement here. It should be noted however that the comparison for AirMSPI is not based on exactly the same points as for SPEX and RSP.

For the comparison between the MAPs (SPEX and RSP) and HSRL-2, we focused on a day with low AOD and a flight leg with high AOD (including measurements with AOD > 1.0) over a prescribed forest fire in Arizona (9 November). For the challenging case of low AOD over land, it was shown that SPEX and RSP are capable of providing accurate retrievals of AOD. For this low AOD case, SPEX showed better comparison against HSRL-2 than RSP.

For the retrievals over the smoke plume also a reasonable agreement in AOD between the MAPs and HSRL-2 was found, despite the fact that the comparison was hampered by large spatial variability in AOD throughout the smoke plume. Besides, a good agreement was found between the MAPs (SPEX, RSP) and HSRL-2 for the aerosol depolarization ratio, which indicates MAPs are capable of retrieving particles sphericity. A reasonable comparison was also found for the aerosol lidar ratio.

For the ALH SPEX retrieved a value that was high (by ∼1.5 km) compared to HSRL-2 while the ALH retrieved from RSP agreed somewhat better with HSRL-2, although it was ∼1 km lower. Here, it should be noted that we do not expect a good ALH retrieval from SPEX airborne, because the shortest wavelength used in the retrieval was 450 nm. For the retrieved microphysical and optical properties of the smoke plume, SPEX and RSP agreed very well with each other and both instruments retrieved smoke properties that were in line with earlier studies.

In this study, 3 polarimeters produced comparable results when using the same algorithm. The exception were the ALH and some coarse mode parameters, which were mainly caused by not having the bands that these parameters were sensitive to: shortwave (410 nm) and SWIR, respectively. For parameters that the instruments were sensitive to, good agreements were found among instruments. Our results corroborate the findings of earlier studies that different combinations of spectral and angular measurements yield a very similar retrieval capability for aerosol properties (Hasekamp and Landgraf, 2007; Wu et al., 2015; Hasekamp et al., 2019)

*Data availability.* The ACEPOL data from MAPs and lidars can be downloaded from the website: https://www-air.larc.nasa.gov/cgi-bin/ArcView/acepol, (registration required). The AERONET data can be downloaded from the website: https://aeronet.gsfc.nasa.gov/. The meteorological NCEP data can be accessed through the website: http://www.cdc.noaa.gov/. The polarimetric retrieval results will be made available on SRON's ftp site.

*Competing interests.* The authors declare that no competing interests are present.

*Acknowledgements.* This work is funded by a NWO/NSO project ACEPOL: Aerosol Characterization from Polarimeter and Lidar under project number ALW-GO/16-09. We thank all the members involved in the ACEPOL campaign. We acknowledge the former Aerosol, Cloud, Ecosystem (ACE) program at NASA's Earth Science Division as a sponsor for ACEPOL flights. We thank the field teams making
5  measurements on the ground as some of those were critical to the measurement accuracy (i.e. vicarious calibration). We also acknowledge the support received by the ground- and aircrew at NASA AFRC in Palmdale and the work performed at the Jet Propulsion Laboratory, California Institute of Technology. We thank the AERONET team and the MERRA-2 team for maintaining the data. NCEP Reanalysis data provided by the NOAA/OAR/ESRL PSD, Boulder, Colorado, USA, from their website at https://www.esrl.noaa.gov/psd/. We would also like to thank the Netherlands Supercomputing Centre (SURFsara) for providing us with the computing facility, the Cartesius cluster. We are
10  very grateful to the editor, Dr Remer, two other reviewers, and Dr Korkin for their reviews and insightful comments.

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

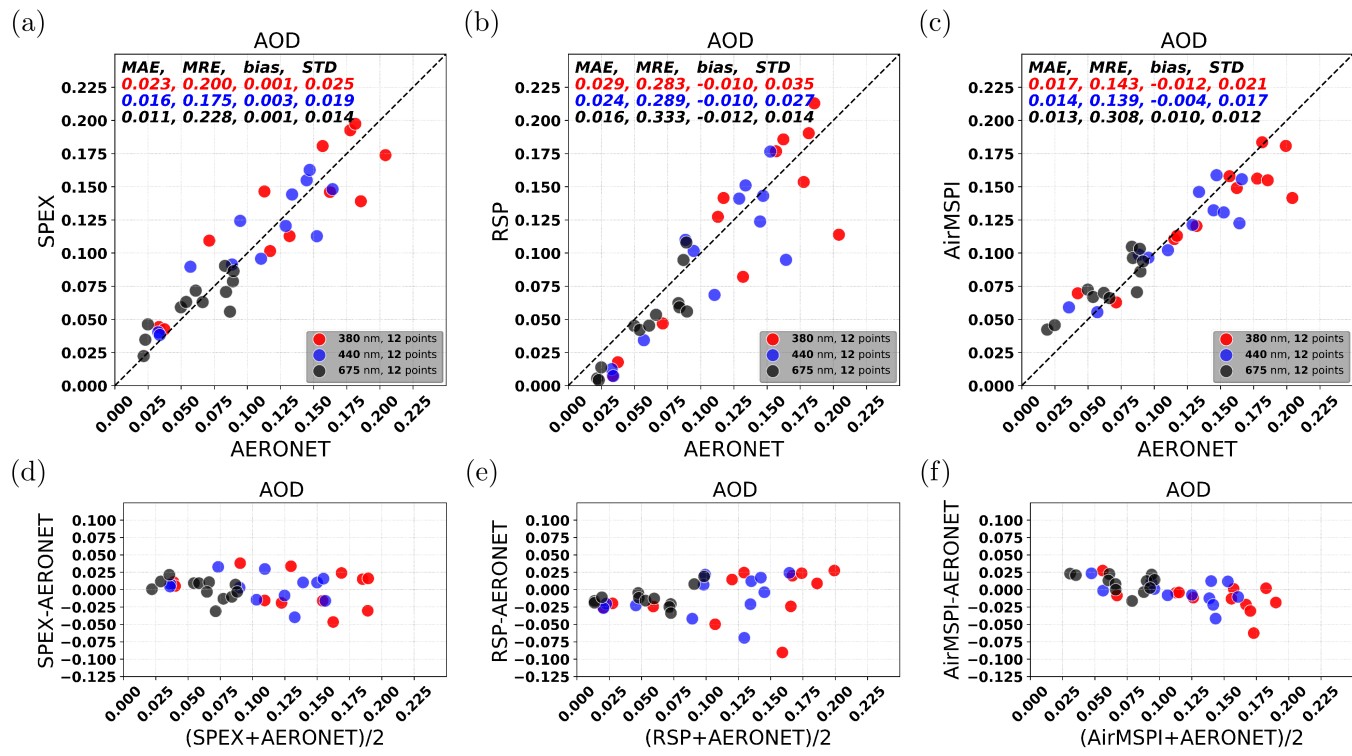

**Figure 1. Comparison with AERONET for AOD (380 nm, 440 nm and 675 nm) among SPEX, RSP, and AirMSPI retrievals.** **(a)**,**(b)**,**(c)** SPEX, RSP, and AirMSPI comparison with AERONET respectively. **(d)**,**(e)**,**(f)** Bland-Altman plots (or difference plots) between SPEX and AERONET, between RSP and AERONET, and between AirMSPI and AERONET, respectively.

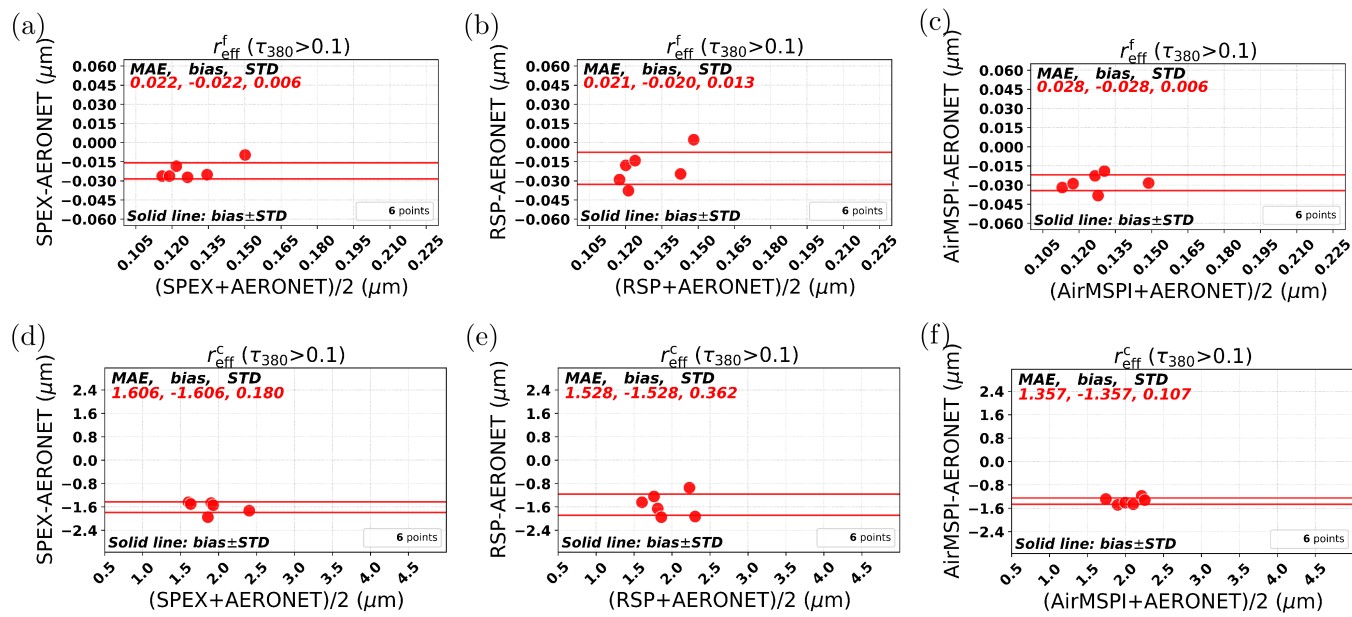

**Figure 2. Comparison with AERONET for the effective radius of the fine and coarse modes ($r_{\mathrm{eff}}^{\mathrm{f}}$ and $r_{\mathrm{eff}}^{\mathrm{c}}$), among SPEX, RSP, and AirMSPI retrievals.** **(a)**,**(b)**,**(c)** Bland-Altman plots for $r_{\mathrm{eff}}^{\mathrm{f}}$ between SPEX and AERONET, between RSP and AERONET, and between AirMSPI and AERONET, respectively. **(d)**,**(e)**,**(f)** Bland-Altman plots for $r_{\mathrm{eff}}^{\mathrm{c}}$ between SPEX and AERONET, between RSP and AERONET, and between AirMSPI and AERONET, respectively.

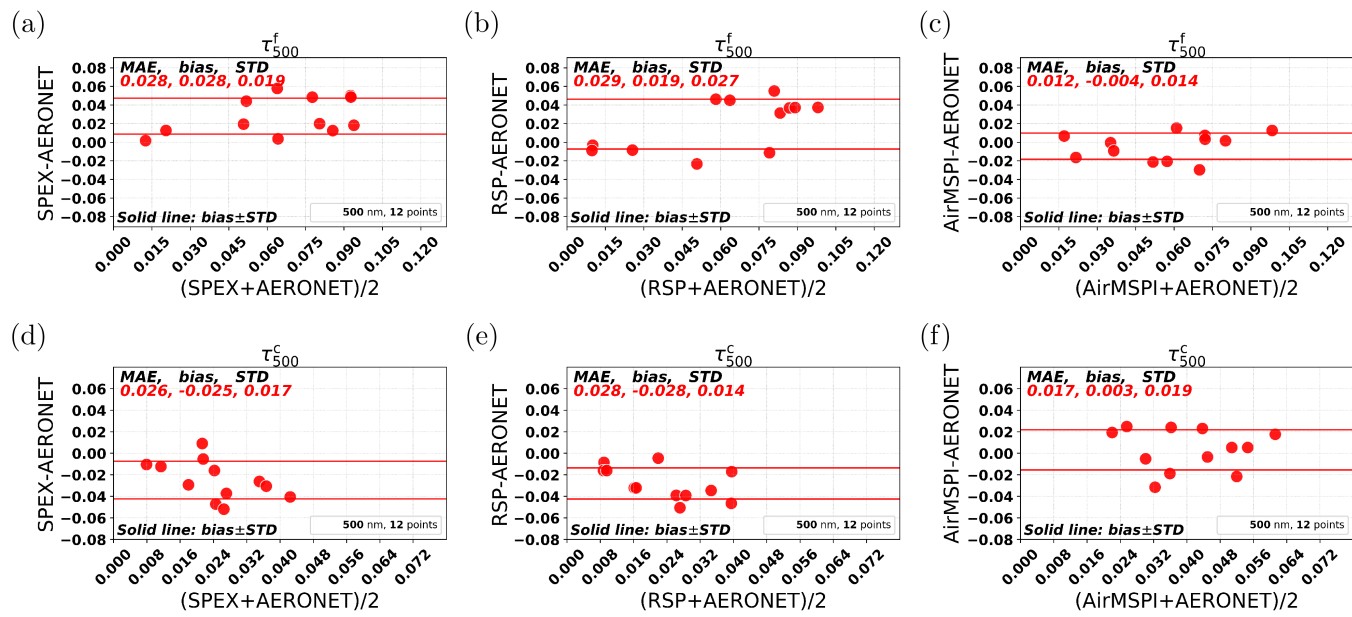

**Figure 3. Comparison with AERONET for the AOD of the fine and coarse modes** ($\tau_{500}^{f}$ **and** $\tau_{500}^{c}$) **among SPEX, RSP, and AirMSPI retrievals.** **(a)**,**(b)**,**(c)** Bland-Altman plots for $\tau_{500}^{f}$ between SPEX and AERONET, between RSP and AERONET, and between AirMSPI and AERONET, respectively. **(d)**,**(e)**,**(f)** Bland-Altman plots for $\tau_{500}^{c}$ between SPEX and AERONET, between RSP and AERONET, and between AirMSPI and AERONET, respectively.

(a)

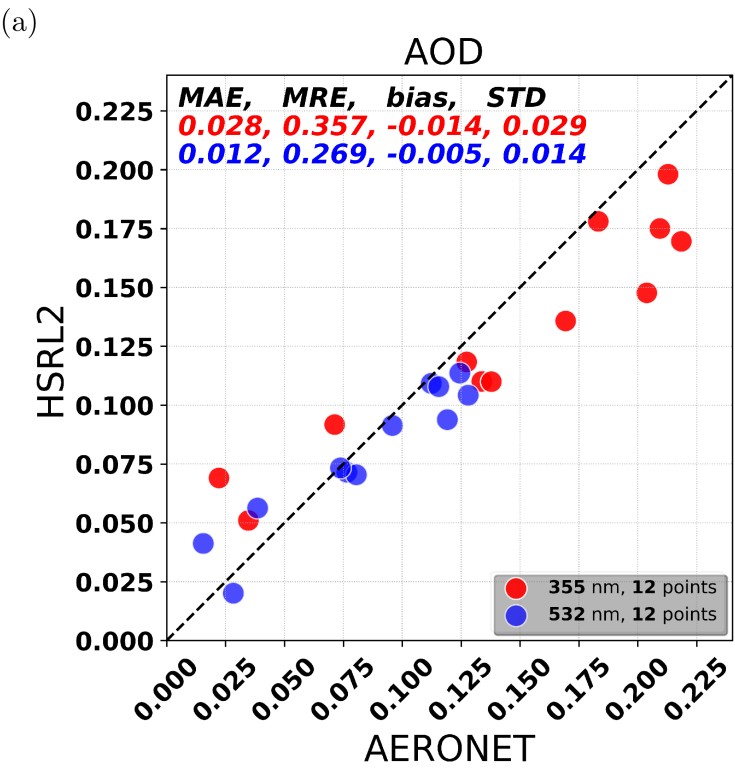

(b)

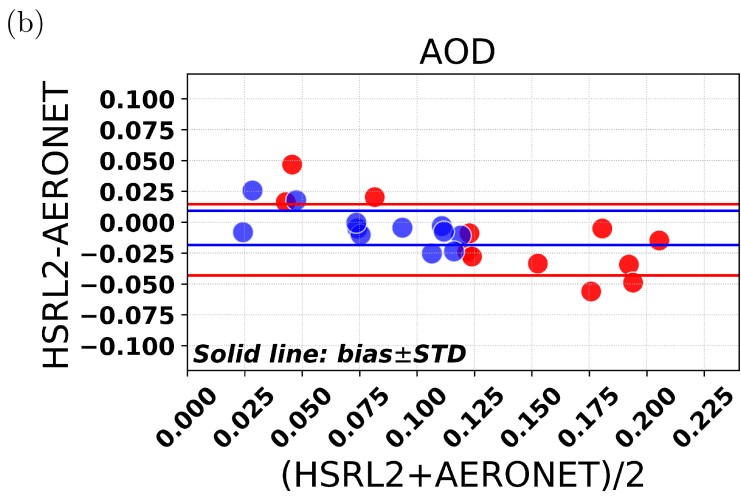

**Figure 4. Comparison between HSRL-2 and AERONET for AOD at 355 nm and 532 nm. (a)** and **(b)** are the scatter plot and the difference plot, respectively.

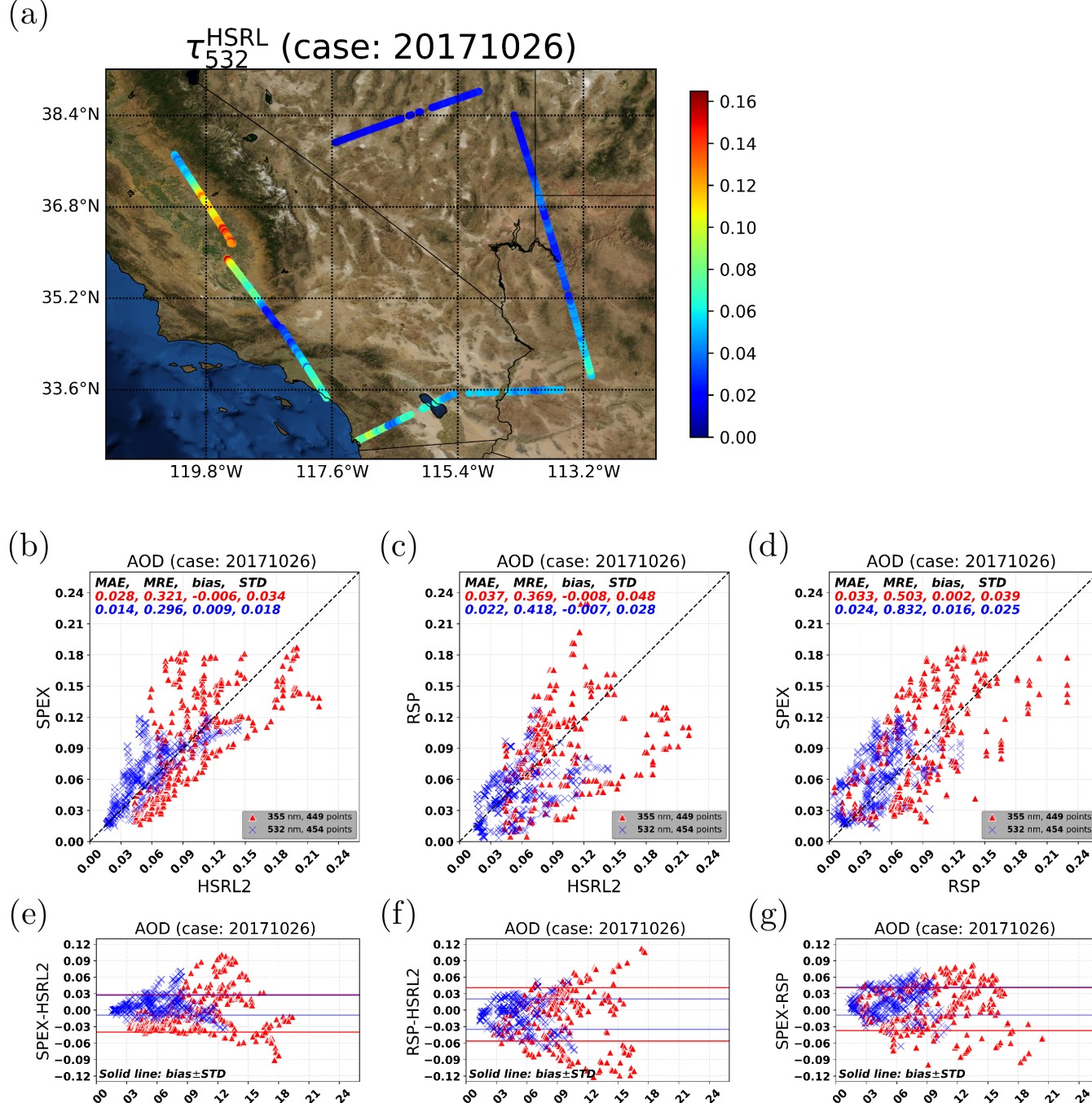

**Figure 5. Comparison with HSRL-2 from 26 Oct 2017 (low AOD case) for AOD (355 nm and 532 nm) between SPEX and RSP retrievals. (a)** HSRL-2 AOD colocation with SPEX and RSP. (The map is generated using python's basemap package and its arcgis image service 'ESRI_Imagery_World_2D'.) **(b)** SPEX AOD comparison with HSRL-2. **(c)** RSP AOD comparison with HSRL-2. **(d)** SPEX AOD comparison with RSP. **(e),(f),(g)** Bland-Altman plots for **(b),(c),(d)**, respectively.

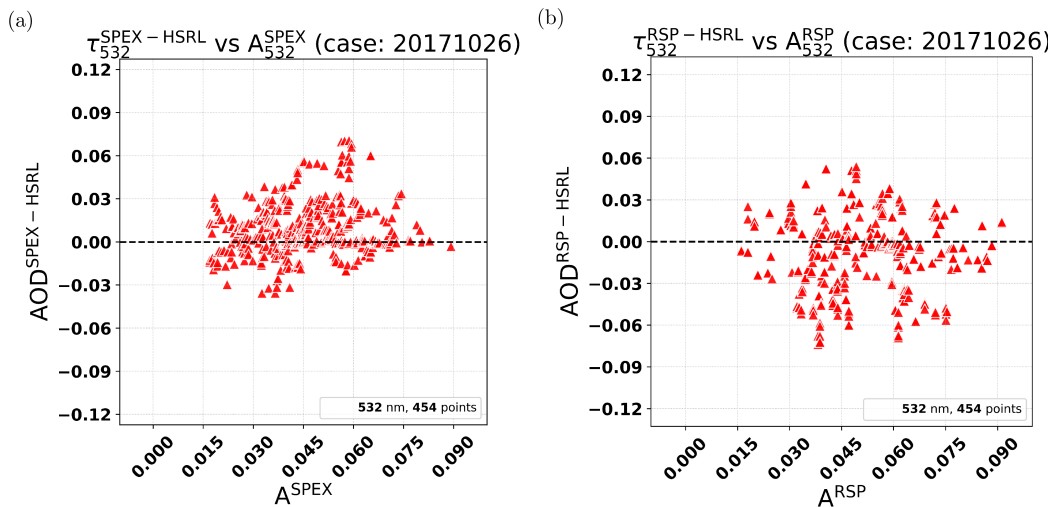

**Figure 6. The AOD difference between MAP and HSRL-2 as function of retrieved BRDF scaling parameter** $A$ **at 532 nm. (a)** AOD difference between SPEX and HSRL-2 versus $A$. **(b)** AOD difference between RSP and HSRL-2 versus $A$.

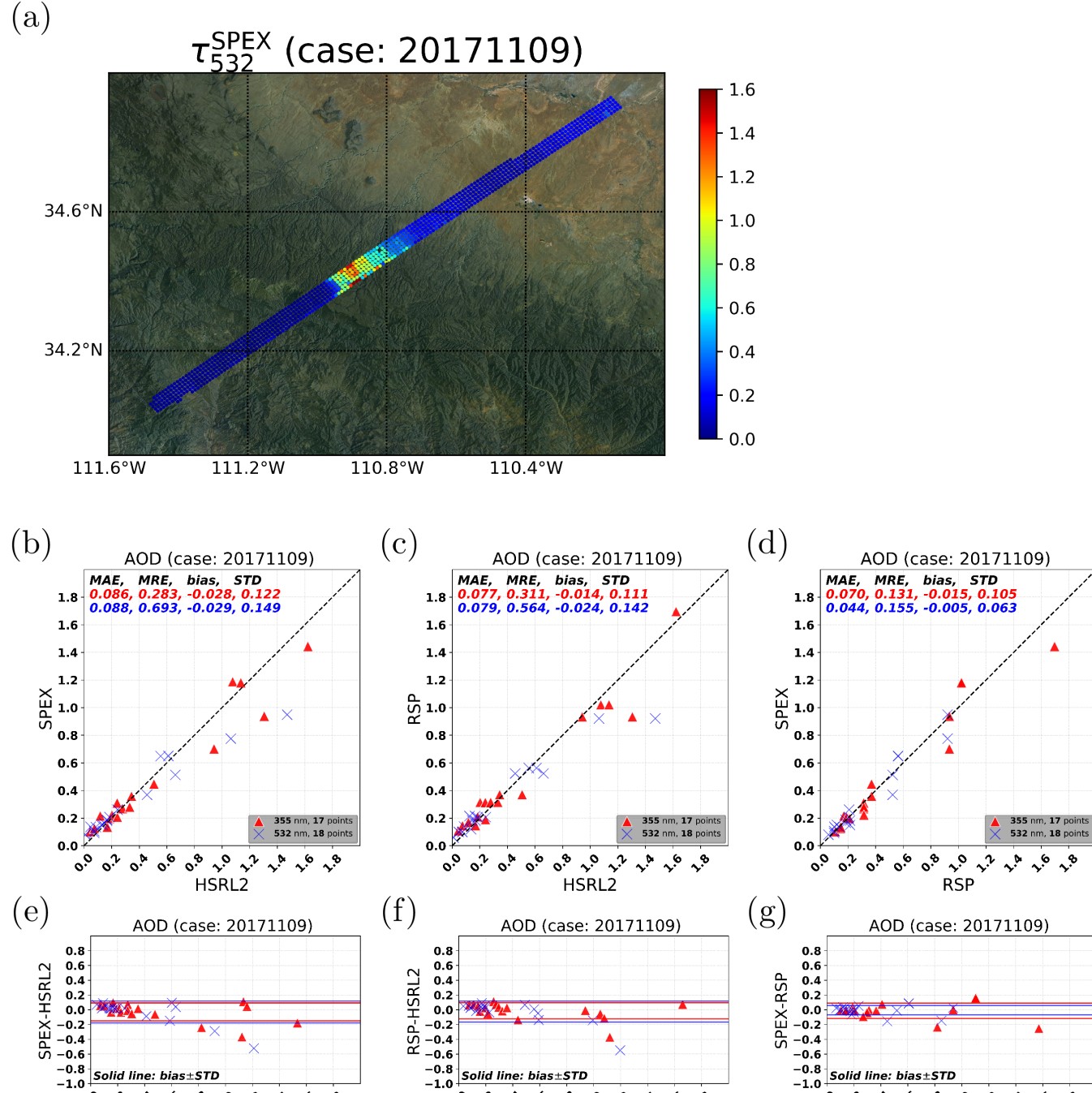

**Figure 7. Comparison with HSRL-2 from 9 Nov 2017 (high AOD smoke case) for AOD (355 nm and 532 nm) between SPEX and RSP retrievals. (a)** SPEX original AOD, i.e., no filter or colocation included. (The map is generated using python's basemap package and its arcgis image service 'ESRI_Imagery_World_2D'.) **(b)** SPEX AOD comparison with HSRL-2. **(c)** RSP AOD comparison with HSRL-2. **(d)** SPEX AOD comparison with RSP. **(e),(f),(g)** Bland-Altman plots for **(b),(c),(d)**, respectively.

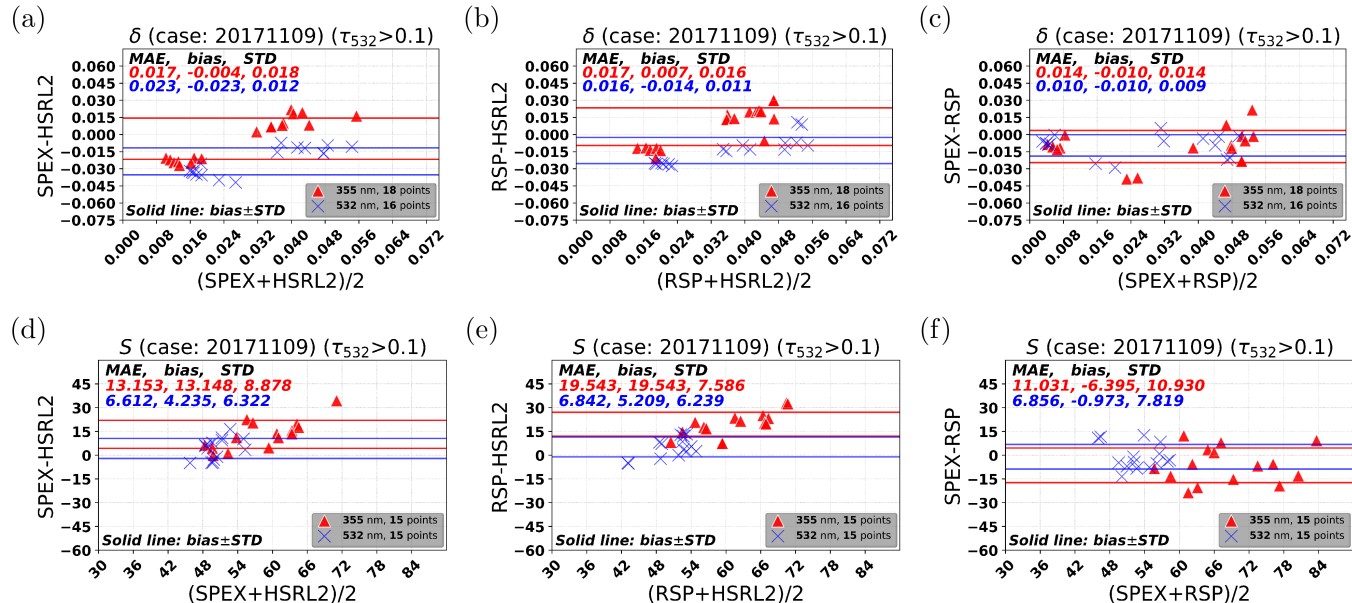

**Figure 8. Comparison with HSRL-2 from 9 Nov 2017 (high AOD smoke case) for the aerosol depolarization ratio ($\delta$) and the aerosol lidar ratio ($S$) between SPEX and RSP retrievals. (a)** SPEX $\delta$ comparison with HSRL-2. **(b)** RSP $\delta$ comparison with HSRL-2. **(c)** SPEX $\delta$ comparison with RSP. **(d)** SPEX $S$ comparison with HSRL-2. **(e)** RSP $S$ comparison with HSRL-2. **(f)** SPEX $S$ comparison with RSP.

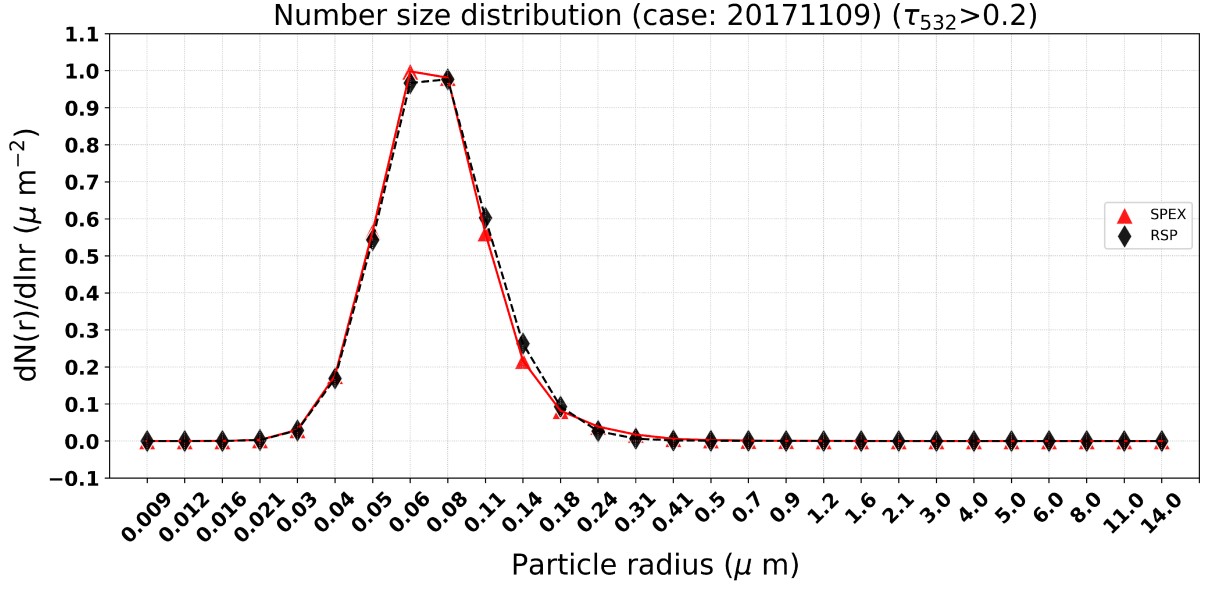

**Figure 9. Number particle size distribution in the smoke plume retrieved from SPEX and RSP.**

**Table 1.** Definition of the effective radius ($r_{\mathrm{eff}}$) and the effective variance ($v_{\mathrm{eff}}$) in the SRON 5-mode retrieval.

|  | Mode 1 | Mode 2 | Mode 3 | Mode 4 | Mode 5 |
|---|---|---|---|---|---|
| $r_{\mathrm{eff}}$ (μm) | 0.094 | 0.163 | 0.282 | 0.882 | 1.759 |
| $v_{\mathrm{eff}}$ | 0.130 | 0.130 | 0.130 | 0.284 | 1.718 |

**Table 2.** Viewing angles and wavelengths used in retrievals among SPEX airborne, RSP, and AirMSPI, and the retrieved parameters from them. Prior values and weighting factors for the state vector are also listed in the table. In this paper, 5-mode retrieval is used, thus $n_{\text{mode}} = 5$. The arrow "→", "←", or "↑" means the same value with the arrow direction. The prior value and weighting factor of aerosol loading for each mode are calculated based on Mie theory using the prior information of AOD from the table (listed in the row of aerosol loading).

| Polarimeters | SPEX airborne | RSP | AirMSPI | | |
|---|---|---|---|---|---|
| Viewing angles | $\pm 56°, \pm 42°, \pm 28°,$ $\pm 14°$, and $0°$ ($n_{\text{vza}}^{\text{spex}} = 9$) | Averaged based on ~150 angles ($n_{\text{vza}}^{\text{rsp}} = 10$) | $\pm 66°, \pm 59°, \pm 48°,$ $\pm 29°$, and $0°$ ($n_{\text{vza}}^{\text{airmspi}} = 9$) in step-and-stare mode | | |
| Wavelengths (radiance) | 450, 460, 470, 480, 490, 500, 510, 520, 530, 540, 550, 565, 580, 600, 670, and 750 nm ($n_{\text{wave}}^{\text{spex}} = 16$) | 410, 469.1, 554.9, 670, and 863.4 nm ($n_{\text{wave}}^{\text{rsp}} = 5$) | 355, 380, 445, 470, 555, 660, and 865 nm ($n_{\text{wave}}^{\text{airmspi}} = 7$) | | |
| Wavelengths (polarization) | ↑ | ↑ | 470, 660, and 865 nm | | |

| | | | | | Prior | Weight |
|---|---|---|---|---|---|---|
| **Retrieved Aerosol properties** | Aerosol loading for each mode | → | $N^j$, $(j = 1, 2, ..., n_{\text{mode}})$ | ← | 0.0001 | $\left(\frac{0.25}{n_{\text{mode}}}\right)^2$ |
| | Fraction of spheres | → | $f_{\text{sphere}}^{\text{c}}$ | ← | 0.5 | 0.25 |
| | Fine mode component coefficient 1 (INORG) | → | $\alpha_1^{\text{f}}$ | ← | 0.95 | $0.1^2$ |
| | Fine mode component coefficient 2 (BC) | → | $\alpha_2^{\text{f}}$ | ← | 0.005 | $0.1^2$ |
| | Coarse mode component coefficient 1 (INORG) | → | $\alpha_1^{\text{c}}$ | ← | 0.95 | $0.1^2$ |
| | Coarse mode component coefficient 2 (DUST) | → | $\alpha_2^{\text{c}}$ | ← | 0.005 | $0.1^2$ |
| | Aerosol layer height (km) | → | ALH | ← | 2.0 | $4.0^2$ |
| **Retrieved Surface properties** | BRDF scaling parameters for wavelength bands | $A(1, 2, \cdots, n_{\text{wave}}^{\text{spex}})$ | $A(1, 2, \cdots, n_{\text{wave}}^{\text{rsp}})$ | $A(1, 2, \cdots, n_{\text{wave}}^{\text{airmspi}})$ | 0.0 | $0.5^2$ |
| | Parameter 1 of RPV model | → | $g$ | ← | -0.09 | $0.5^2$ |
| | Parameter 2 of RPV model | → | $k$ | ← | 0.80 | $0.5^2$ |
| | Scaling parameter for polarized reflectance | → | $B$ | ← | 4.0 | $2.0^2$ |

**Table 3.** Median and standard deviation (STD) properties of the smoke plume from SPEX and RSP when AOD > 0.2 at 532 nm.

| | SPEX | | RSP | |
|---|---|---|---|---|
| | Median | STD | Median | STD |
| Fine mode real part of refractive index ($m_{\mathrm{r},532}^{\mathrm{f}}$) | 1.579 | 0.019 | 1.556 | 0.059 |
| Fine mode imaginary part of refractive index ($m_{\mathrm{i},532}^{\mathrm{f}}$) | 0.038 | 0.011 | 0.036 | 0.013 |
| Fine mode effective radius ($r_{\mathrm{eff}}^{\mathrm{f}}$) | 0.116 | 0.004 | 0.119 | 0.007 |
| Fine mode AOD ($\tau_{532}^{\mathrm{f}}$) | 0.554 | 0.238 | 0.509 | 0.231 |
| Coarse mode AOD ($\tau_{532}^{\mathrm{c}}$) | 0.016 | 0.011 | 0.040 | 0.029 |
| Aerosol layer height (ALH) (km) | 4.417 | 1.148 | 1.585 | 1.588 |
| SSA ($\omega_{532}$) | 0.815 | 0.044 | 0.829 | 0.044 |
| Fraction of spherical particles ($f_{\mathrm{sphere}}$) | 0.989 | 0.149 | 0.846 | 0.133 |