# Peer review of "Aerosol retrievals from different polarimeters during the ACEPOL campaign using a common retrieval algorithm"

_Atmospheric Measurement Techniques, 2019_

## Short Comment (SC1) · 13 Aug 2019

The 'alpha \* Rpol' part in Eq.(1), p.4, seems unclear to me. Please provide explicit form for the Rpol matrix like it is done for the RPV-part. Reference to the Maignan et al, 2009 paper gives little help:

1) Sections 3.2, 3.3, and 3.5 in Maignan et al. 2009 discuss different models - which particular one was used in the paper under review?

2) As far as i know, Maignan et al. 2009 does not define the Mueller matrix of the surface - only BPDF, which is based on the F12=F21=Fp element of the Fresnel matrix. To get the Mueller matrix, shall one compute matrix exponential of the Fresnel matrix, or, vice versa, create a matrix of "scalar" exponents of elements of the Fresnel matrix?

3) The complete 4x4 (or reduced 3x3 if V is ignored) Mueller matrix of the surface is

required only to simulate the surface reflection of diffuse radiation (including multiple bouncing of light between the atmosphere and surface). How strong and important is that effect for polarization components?

4) Is 'alpha' in Eq.(1) band-dependent?

Thank you!

---

## Referee Comment (RC1) · Lorraine Remer (Referee) · 20 Aug 2019

This manuscript presents a straightforward, well-executed and well-written comparison of aerosol retrievals from three airborne multi-angle polarimeters, an airborne HSRL (lidar) and AERONET. The polarimeters and HSRL are simultaneously flying on the NASA ER-2 during ACEPOL, and all data shown are collocated in time and space, or are clearly described as being exceptions. The same algorithm is applied to the three polarimeters so that differences should be instrumental not algorithmic. I have nothing to criticize in this study or presentation, and offer only minor comments, as suggestions, below.

Before that, I will reveal myself. This is Lorraine Remer writing.

1. General comment. I understand that one of the purposes of this work is to determine expected uncertainty on the retrievals for the polarimeters. But AERONET and HSRL

should already have documented expected uncertainty. It certainly would be helpful to indicate on the figures what is the expected uncertainty of the known sensors. We see values for MAE, bias, etc., but do not know how to put these values into context. If we knew AERONET uncertainty for that parameter, for example, context could be established.

2. General comment. This is a corollary to (1). AERONET AOD has very small uncertainty, but AERONET retrieved products and these include the SDA products have larger error bars. The goal in comparing polarimeter retrievals to these other retrievals is comparison, not validation. This was not explicitly stated in the paper.

3. General comment. I see in the description of the different data sets mitigating strategies for inhomogeneity for registering the different angular views. Does this include topographical variation?

4. Page 9. Last paragraph that begins with "As with the extinction products", I'm a little unclear on what is being said here. "HSRL method" is when HSRL measures extinction. "assumed lidar ratio" is when it does not. The HSRL method is not available in many situations during ACEPOL, so the lidar information is going to come to us like an old-fashioned backscattering lidar with an assumed lidar ratio. It's not clear why the HSRL method is going to be unavailable. Then here it seems to imply that there is going to be a choice between the two methods, not that the HSRL method is unavailable, but that both are available. And then it says that the assumed lidar ratio method is actually BETTER than the HSRL method at low loading. This is because one measures its uncertainty in a relative sense and the other in an absolute sense. The fact that the assumed lidar ratio can be better than the HSRL method is very strange to me. Did I understand this paragraph correctly?

5. Section 3.5. AERONET section. Level 1.5 is cloud cleared, but not quality controlled. Also be aware that fine and coarse as defined by both the almucantar retrievals and the SDA methods are going to be different than defining fine and coarse by specific

modes as is done in the polarimeter retrieval (Table 1). This may introduce differences in your comparisons. It did with the MODIS Dark Target over ocean retrieval.

6. Page 11-12. Discussion of comparison of effective radius against AERONET. Perhaps AERONET is wrong here? This is retrieval vs. retrieval, not retrieval vs. truth. And the loading is extremely low. I would think that everybody is running on fumes here. This applies to fine mode, but especially to coarse mode. Nobody has SWIR to really nail coarse mode. And AERONET's definition of fine and coarse modes, and their respective effective radii, are defined differently than the five modes in Table 1.

7. Figure 3. If I'm interpreting these plots correctly... The MAP retrievals can be very different from AERONET. For example, RSP has differences of -0.04 where the (AERONET + RSP)/2 = 0.025. This means that RSP retrieved tau_c of 0.005 and AERONET 0.045. In absolute terms that's not a lot, but in terms of relative contributions of the coarse mode to the total AOD it is a lot. Is it within expected error of the AERONET retrieval? It would be very helpful to have some context for the magnitude of the differences.

8. Section 4.2.1 These comparisons are all with "assumed lidar ratio". Are these the only days with collocations? If there is a choice between assumed lidar ratio and HSRL method, how does the HSRL method compare?

9. Section 4.2.2. I grew up in Los Angeles and the Central Valley, so I know this territory well, but not everybody does. Maybe use "east" and "west" without place names, or annotate the image.

10. Final sentence of Section 4.2.2. " The differences from the direct comparison between SPEX and RSP are somewhat larger than those from individual comparisons with HSRL-2 of SPEX and RSP, respectively. This suggests that the differences with HSRL-2 are not caused by common assumptions in the SPEX and RSP retrievals, but are rather caused by errors that are specific to each MAP." I don't follow the logic.

11. Section 4.2.3. page 13. Lines 15-17. "It should be noted that the smoke plume exhibits large spatial variation so part of the MAP-lidar differences can be attributed to the fact that different instruments see a slightly different part of the smoke plume." What about different angles from the same instrument seeing different parts of the smoke, or what if the smoke changes between the fore and aft angles are measured? What happens to the retrieval? It would be really nice to have a quantitative sense of how variable that plume is. Could we see a spatial plot of the smoke retrievals or at least have stdev on the parameters shown in Table 2.

12. Page 13. Lines 25-26. "Our explanation for this, is that at high AOD the measured radiance and DoLP are less affected by the co-registration errors between viewing angles than for low AOD." How could this be? The evolving, hetereogeneous smoke plume has to be more difficult to co-register between angles than the unmoving ground.

13. Page 13. Lines 33-34. On the other hand, I think this is a really good explanation: "A possible explanation for the difference could be the simplified description of non-spherical particles in our retrieval approach. "

14. Figure 7d-f. Are lidar ratios here retrieved via HSRL method, or assumed? If assumed, does these figures make any sense. If retrieved, then why not use retrieved throughout the paper? Or show that they are worse than assumed. This whole retrieved vs. assumed lidar ratio choice never sat well with me throughout the manuscript.

15. Table 2. Maybe show stdev along with mean? Or show spatial distribution if any of these properties are varying downwind?

16. Page 14. Line 13. "the latter value is closer to the ALH derived from HSRL-2 (2.64 km)." Sure slightly closer, but still 1 km off. Not that much different from SPEX.

17. Page 14. Line 14. The explanation of ALH being difficult to retrieve without UV might be elaborated on a little here.

18. Finally... don't you want to state a conclusion? What is the overarching thing

you have learned? If this were my paper I would conclude that the 3 polarimeters are producing comparable results when forced through the same algorithm. The exception being aerosol layer height and perhaps some coarse mode parameters, which suffer from not having the bands that these parameters are sensitive to: shortwave (410 nm) and SWIR, respectively. So when there is no sensitivity, the retrieval becomes a random number generator. But for parameters that the instruments are sensitive to, there is little difference between instruments. It is still TBD whether algorithmic differences are going to matter. But it is not my paper. The authors can choose to write a conclusion of their choice. Or not.

---

## Referee Comment (RC2) · Anonymous Referee #2 · 26 Aug 2019

This paper presents aerosol retrievals from polarized measurements of three different airborne instruments in the ACEPOL campaign. The retrieval results are compared with AERONET inversion products and the lidar onboard the same flight, HSRL2 observations. A good agreement of both AOD and aerosol microphysical properties is found among these observations. Furthermore, aerosol retrievals in two special cases with low and high AOD (smoke plume) are shown. I think this is a good study and suggested to be published after some revisions.

**General comments**

1. When the authors introduce the SRON multimode retrieval algorithm in section 2.1, no aerosol size distribution parameters are included in the state vector. However, in the retrieval results, effective radius of fine and coarse mode particles are shown. Although the calculation of fine and coarse mode effective radius is presented in section 2.2, the retrieved aerosol parameters related to size parameters are not clear.

2. As defined in the manuscript, the $\chi^2$ used to decide retrieval convergence is different for different instruments. For example, for AirMSPI, observed intensities in 8 bands and DoLP in 3 bands are used in the retrieval, while radiance and DoLP at 16 wavelengths for SPEX are used. Although $\chi^2$ is defined as a mean value of total number of measurements, the ratios $\frac{(F_i-y_i)^2}{S^y(i,i)}$ in Eq. 3 for radiance and DoLP may have different scales. Therefore, if different numbers of radiances and DoLP are used even though two instruments have the same total number of measurements, the $\chi^2$ may differ a lot. Does this problem affect the retrieval results between different instruments? Do they use the same threshold $\chi_{max}^2$?

3. The retrieval results of 3 different instruments are compared in this manuscript, but only some statistical parameters, such as MAE, bias and STD are presented. Are there any conclusions or suggestions about the measurements (radiance or DoLP) at which wavelengths are combined better for aerosol retrieval? Or are different numbers of multi-angle measurements affect aerosol retrievals a lot? I think more similar common summaries could attract audiences.

4. In the state vector, aerosol column numbers and microphysical properties are included, thus the AOD in the retrieval at different wavelengths are calculated from retrieved column numbers and other parameters. I'm a little confused that why the authors use different wavelengths when compare total AOD and fine and coarse modes AOD (Figure 1 and Figure 3). If the same wavelengths are used, the retrieval performance of fine, coarse mode AOD and total AOD can also be evaluated.

5. The surface reflectance parameters are retrieved simultaneously with aerosol

properties in the algorithm. How is the performance of surface reflectance retrieval in the campaign? Are the accuracies of retrieved aerosol properties related to surface reflectance?

6. The retrieval accuracy of fine and coarse mode AOD depend on the retrieved aerosol microphysical properties. If the dependence of the retrieval bias of $\tau^f$ and $\tau^c$ on the accuracy of retrieved $r_{eff}$ or refractive index is shown, it will be interesting and beneficial for distinguishing aerosol types.

**Specific comments**

1. In the introduction part, the third paragraph in page 2 indicates that combining both intensity and polarization measurements at multiple viewing angles is beneficial for aerosol retrieval. However, this paragraph is too short and simple. This is the most important feature of 3 MAPs used in this manuscript to do retrieval. I think more theoretical foundation and how previous studies use these information could be added.

2. The paragraph at page 3 line 6-10 has little relationship with this study. I believe the authors could delete or short this paragraph and combine it with last paragraph.

3. When giving the information of ACEPOL campaign in the introduction, the information about the altitude aircraft flying is suggested to be provided due to the retrieval of ALH, especially at smoke plume case whose ALH is always high.

4. At page 4 line 20, the meaning of k in the equation is not explained.

5. At page 11 line 18-19, the authors present "the MAE gets smaller with increasing wavelengths, which is mainly caused by the fact that AOD value itself decreases

with wavelength". Some other parameters such as mean relative error (MRE) or root mean squire error (RMSE) could remove this effect and are recommended to be compared.

6. The sentences at line 22-23 and line 30-31 in page 11 present the same thing.

7. At page 13 line 1-2, "for low AOD the effect of the surface on the measured radiances is larger than for SPEX airborne" is presented. I'm a little confused why.

8. At page 14 line 13-14, the authors explained that the shortest wavelength for SPEX is 450 nm and not suitable for ALH retrieval. Do you mean the shorter wavelengths such as UV band benefit ALH retrieval? More clear and straight forward sentences are suggested to be used. Moreover, this explanation for ALH retrieval is too simple and this may be only one of many reasons. I believe reading more related papers about ALH retrieval could help the authors explain this problem more clearly and deeply.

9. Some sentences in this manuscript are a little complex and confused, especially in section 1 and section 4. More concise sentences are recommended.

---

## Referee Comment (RC3) · Anonymous Referee #3 · 4 Sep 2019

This article by Fu et al presents aerosol retrievals from three polarimeters (SPEX, RSP, and AirMSPI) deployed in the 2017 NASA-SRON ACEPOL aircraft campaign with the SRON aerosol inversion algorithm developed by Dr. Hasekamp's research group. The retrieval results are evaluated through comparisons with measurements from HSRL2 flown on the same ER-2 aircraft and available AERONET observations, as well as through the inter comparisons of retrievals among these polarimeters. Overall, this is a well-written and proficiently organized article. The results are sound and properly discussed. I agree with most of the comments given by Dr. Remer and other reviewers. Below I have a few additional comments that the authors should consider to address.

General comments:

1. Although the paper focuses on aerosol retrievals, surface is an important component in the retrieval process and is included in the state vector. A good characteri-

zation of surface reflectance can significantly affect the retrieval accuracy of aerosol properties, which is especially true when aerosol loading is small (such as of the most ACEPOL cases). So, as a reader I would like to see some retrieval results for surface BRDF/BPDF properties and how the retrievals behave between different polarimeters.

2. The retrieval algorithm needs some more clarification in a few aspects of the radiative transfer calculations and the inversion configurations. These include: (i) which radiative transfer model and what are the relevant assumptions (such gas absorptions, Rayleigh scatterings, etc) in the radiative transfer assumptions? (ii) How the first guess of the state vector is defined? While the first guesses for aerosol parameters are mostly given, the paper mentions nothing about prior values for surface BRDF/BPDF parameters. (iii) It is not clear how the aerosol refractive index are treated, although it is mentioned to use the D'Almeida et al (1991) database. (iv) It is also not clear about how the weighting matrix (W) in the cost function is defined, as well as the threshold for the goodness of fit. Please refer to the relevant specific comments below for more details.

3. By reading the title of the article (Aerosol retrievals from the ACEPOL campaign), I would expect to see aerosol retrievals from different polarimeters and from their respective aerosol products. Are there any aerosol products available from the ACEPOL campaign with other existing retrieval algorithms? If yes, it would be more helpful to compare the aerosol retrievals from different algorithms. Such a comparison may also explain the consistent biases in the retrieved aerosol size (Figure 2), depolarization ratio and lidar ratio (Figure 7). Otherwise, I would suggest to make the article title more specific, for instance, by adding "using the SRON algorithm".

Specific comments

1. Page 4, first paragraph of section 2.1. Description about aerosol refractive index is too brief. Please clarify: (i) at which relative humidity (RH) is assumed for the D'Almeida et (1991) database, or a dynamic RH relationship is considered with ancillary meteorological data? This is important as the inorganic aerosols are strongly hygroscopic. (ii) How the coefficients are defined for combining the aerosol species? In terms of volume concentrations? (iii) Are the different aerosol species internally or externally mixed in the calculation of modal refractive index? In addition, it would be helpful if the refractive indices used in this study being provided in a supplemental document.

2. Page 5, line 6. Please give the explicit expression for R(G).

3. Section 2.1. The number of elements in state vector for different sensors would be different because of the different number of spectral bands. I would recommend include a table to list the detailed elements (and numbers) of the retrieved parameters for individual polarimeters. Correspondingly, the selected bands and number of angles for each observation set (as described in section 3.1-3.3) can also be listed in the same table. This will give the reader a clearer picture about the retrieval configuration for different sensors.

4. Section 2.1. It is not mentioned in algorithm description about: (i) what radiative transfer model is used and how many layers of atmosphere is assumed; (ii) how the gas absorption are treated; (iii) How the Rayleigh scattering are calculated. Please clarify.

5. Page 5, Equation (2). Please clarify how the wright matrix (W) is defined to regulate the ranges of individual state parameters.

6. Page 5, Equation (2). It is not clear how the prior state vector is defined for surface parameters. Please clarify.

7. Page 5, line 15. It is mentioned here "Stokes parameters I, Q, U at the top of the atmosphere" are simulated, but it is not clear what is the TOA altitude as defined. Moreover, the ACEPOL measurements are taken at an altitude of the ER-2 flights. The radiative transfer model should simulate the radiances as observed at the flight level. Please justify.

8. Page 5, line 29. Is a constant threshold for Kai-Square used for all retrievals across different instruments? Please clarify.

9. Page 6, Equation (4). The symbol "G" is already used in equation (1) to denote hot-spot geometry factor. A different symbol should be used to avoid ambiguity.

10. Page 6, Equation (7). Are there any references for calculating the columnar de-polarization ratio in this way? I recall some studies (sorry I couldn't find the paper) used layer extinction coefficient (rather than backscatter coefficient) as the weighting parameter.

11. Page 9, line 24. Do you meant to "Where the HSRL method is NOT available for the extinction products . . .."

12. Page 11, line 32. It seems the effective radius for coarse modes 4 and 5 are much smaller than the AERONET climatology as reported in Dubovik et al (2002). So why not define a large effective radius values for these two modes.

Reference: Dubovik, O. et al (2002), Variability of Absorption and Optical Properties of Key Aerosol Types Observed in Worldwide Locations, Journal of the Atmospheric Sciences, 59(3), 590-608.

13. Figure 28. Authors may consider to replace the background of Figure 28a with a true color image of the smoke plume. I have seen such a figure from AirHARP gallery. It would be even better if a retrieved AOD map for the smoke plume is presented here.

14. Page 13, line 23-24. It is mentioned here the smoke plume has large spatial variability that may contribute to the retrieval uncertainty. The suggestion above (#13) would at least give a visual expression how large the spatial variability is. In addition, the MAP algorithm would have challenge to retrieve AOD as different view angles see different location (thus AOD) of the elevated plume due to the parallax displacement. Can the authors provide some insights on how to addressing this challenge in the retrieval?

15. Finally, I would like to see a figure of retrieved particle size distribution for the smoke case, which would help interpreting the retrieval results listed in Table 2.

---

## Author Comment (AC1) · 6 Nov 2019

**Reply to comments**:

1. *The 'alpha \* Rpol' part in Eq.(1), p.4, seems unclear to me. Please provide explicit form for the Rpol matrix like it is done for the RPV-part. Reference to the Maignan et al, 2009 paper gives little help:*
   *1) Sections 3.2, 3.3, and 3.5 in Maignan et al. 2009 discuss different models - which particular one was used in the paper under review? 2) As far as i know, Maignan et al. 2009 does not define the Mueller matrix of the surface - only BPDF, which is based on the F12=F21=Fp element of the Fresnel matrix. To get the Mueller matrix, shall one compute matrix exponential of the Fresnel matrix, or, vice versa, create a matrix of "scalar" exponents of elements of the Fresnel matrix? 3) The complete 4x4 (or reduced 3x3 if V is ignored) Mueller matrix of the surface is required only to simulate the surface reflection of diffuse radiation (including multiple bouncing of light between the atmosphere and surface). How strong and important is that effect for polarization components? 4) Is 'alpha' in Eq.(1) band-dependent?*

   Response:
   We have added the expression (Eq.(2)) and the detailed description for $\mathbf{R}_{\mathrm{pol}}$ in the revised manuscript.

---

## Author Comment (AC2) · 6 Nov 2019

**Reply to comments**:

1. *General comment. I understand that one of the purposes of this work is to determine expected uncertainty on the retrievals for the polarimeters. But AERONET and HSRL should already have documented expected uncertainty. It certainly would be helpful to indicate on the figures what is the expected uncertainty of the known sensors. We see values for MAE, bias, etc., but do not know how to put these values into context. If we knew AERONET uncertainty for that parameter, for example, context could be established.*

   Response:
   We agree. We added a phrase to the paper with respect to the expected AERONET AOD uncertainty in Sect. 3.5:
   " The uncertainty on AERONET AOD is 0.01 for mid-visible wavelengths and 0.03 for UV wavelengths (Eck et al., 1999) and is dominated by a calibration (systematic) error.   "

   For the expected uncertainty in HSRL-2, we refer to the comparison between HSRL-2 and AERONET. We added the phrase in Sect. 4.2.1: " The bias between HSRL-2 and AERONET is within the AERONET uncertainty. The random differences, with standard deviation 0.029 at 380 nm and 0.014 at 532 nm are most likely due to HSRL-2 uncertainties.   "

2. *General comment. This is a corollary to (1). AERONET AOD has very small uncertainty, but AERONET retrieved products and these include the SDA products have larger error bars. The goal in comparing polarimeter retrievals to these other retrievals is comparison, not validation. This was not explicitly stated in the paper.*

   Response:
   We agree and explicitly state this to the paper in Sect. 3.5:
   " The multispectral aerosol optical depth (AOD) from the MAP and lidar retrievals is validated  with AERONET (AErosol RObotic NETwork) level 1.5 data (Holben et al., 2001) (version 3.0). The data are cloud cleared. The uncertainty on AERONET AOD is 0.01 for mid-visible wavelengths and 0.03 for UV wavelengths (Eck et al., 1999) and is dominated by a calibration (systematic) error.   The effective radius for fine and coarse modes are compared  with AERONET level 1.5 Almucantar Retrieval Inversion Products (Dubovik et al., 2002). The AOD of fine and coarse modes are compared  with AERONET level 1.5 spectral de-convolution algorithm (SDA) data (O'Neill et al., 2003). It should be noted that the inversion- and SDA products are quite uncertain themselves at low AOD so the comparison to these products should not be considered a validation.   "

3. *General comment. I see in the description of the different data sets mitigating strategies for inhomogeneity for registering the different angular views. Does this include topographical variation?*

Response:

Yes, this does include topographical variation.

4. *Page 9. Last paragraph that begins with "As with the extinction products", I'm a little unclear on what is being said here. "HSRL method" is when HSRL measures extinction. "assumed lidar ratio" is when it does not. The HSRL method is not available in many situations during ACEPOL, so the lidar information is going to come to us like an old-fashioned backscattering lidar with an assumed lidar ratio. It's not clear why the HSRL method is going to be unavailable. Then here it seems to imply that there is going to be a choice between the two methods, not that the HSRL method is unavailable, but that both are available. And then it says that the assumed lidar ratio method is actually BETTER than the HSRL method at low loading. This is because one measures its uncertainty in a relative sense and the other in an absolute sense. The fact that the assumed lidar ratio can be better than the HSRL method is very strange to me. Did I understand this paragraph correctly?*

Response:

Thanks for pointing out this point. Actually the statement "For ACEPOL, the extinction, AOD, and lidar ratio from the HSRL methodology are not available for many ground pixels." was not accurate (which was based on the old version HSRL-2 data). We have removed this statement and re-wrote that part to avoid confusion in Sect. 3.4: " For ACEPOL, the extinction products from the HSRL method are reported at 150 m vertical resolution and at temporal resolution of 60 s generally and 10 s. Additionally, the aerosol extinction products at 355 nm and 532 nm are also provided based on the aerosol backscatter and an assumed lidar ratio of 40 sr, and reported at the backscatter resolution.

Similarly, the AOD is reported from the standard HSRL approach and also the AOD calculated using assumed lidar ratio is provided. "

Yes, we have two AOD products as two choices. For the low AOD case the AOD from the assumed lidar ratio is better than the HSRL method. This is because for low AOD, both approaches are difficult, but the AOD from the assumed lidar ratio is expected to be smaller (given that the AOD is small) than the systematic uncertainty $\sim$0.05 from the HSRL method for ACEPOL.

5. *Section 3.5. AERONET section. Level 1.5 is cloud cleared, but not quality controlled. Also be aware that fine and coarse as defined by both the almucantar retrievals and the*

*SDA methods are going to be different than defining fine and coarse by specific modes as is done in the polarimeter retrieval (Table 1). This may introduce differences in your comparisons. It did with the MODIS Dark Target over ocean retrieval.*

Response:

Thanks. We added this point to the paper in Sect. 4.1:

" Also, it should be noted that the "fine" and "coarse" as defined by the Almucantar retrievals are different with defining "fine" and "coarse" by specific modes as shown in Table 1. This may introduce differences in the comparisons. "

6. *Page 11-12. Discussion of comparison of effective radius against AERONET. Perhaps AERONET is wrong here? This is retrieval vs. retrieval, not retrieval vs. truth. And the loading is extremely low. I would think that everybody is running on fumes here. This applies to fine mode, but especially to coarse mode. Nobody has SWIR to really nail coarse mode. And AERONET's definition of fine and coarse modes, and their respective effective radii, are defined differently than the five modes in Table 1.*

Response:

We agree and included this aspect to the paper in Sect. 4.1:

" It is important to note that for the low AOD values encountered during ACEPOL, the AERONET retrieved fine and coarse mode AOD and effective radius are very uncertain themselves. Therefore, this comparison should not be interpreted as "retrieval versus truth" but rather as "retrieval versus retrieval". "

7. *Figure 3. If I'm interpreting these plots correctly... The MAP retrievals can be very different from AERONET. For example, RSP has differences of -0.04 where the (AERONET + RSP)/2 = 0.025. This means that RSP retrieved tau_c of 0.005 and AERONET 0.045. In absolute terms that's not a lot, but in terms of relative contributions of the coarse mode to the total AOD it is a lot. Is it within expected error of the AERONET retrieval? It would be very helpful to have some context for the magnitude of the differences.*

Response:

We added some context for the magnitude to the paper in Sect. 4.1:

" The comparison shows a MAE of 0.028, 0.029, and 0.012 for SPEX, RSP, and airMSPI, respectively for $\tau^{\mathrm{f}}$ and 0.026, 0.028, 0.017 for $\tau^{\mathrm{c}}$. The bias is 0.028, 0.019 and 0.004 for $\tau^{\mathrm{f}}$ and 0.025, 0.028, and 0.003 for $\tau^{\mathrm{c}}$. So, SPEX and RSP have an overestimation of the fine mode and an underestimation of the coarse mode, compared to AERONET SDA product. Although these biases are large in a relative sense (given the low AOD, especially for

the coarse mode), they are within the expected error from the AERONET SDA product. AirMSPI compares better to the AERONET SDA product than SPEX airborne and RSP. "

8. *Section 4.2.1 These comparisons are all with "assumed lidar ratio". Are these the only days with collocations? If there is a choice between assumed lidar ratio and HSRL method, how does the HSRL method compare?*

Response:
Indeed all HSRL2-AERONET collocations are included. We have explicitly mentioned in Sect 3.4, the comparisons in Figure 4 are with the assumed lidar ratio. For these low AOD cases, the comparison against AERONET for HSRL AOD from the HSRL method is worse than from the assumed lidar ratio. The reason has been explained in Sect 3.4.

9. *Section 4.2.2. I grew up in Los Angeles and the Central Valley, so I know this territory well, but not everybody does. Maybe use "east" and "west" without place names, or annotate the image.*

Response:
Thanks. We changed them to the paper in Sect. 4.2.2:
" From this figure it follows that there were very low AOD values for the eastern part of the scene and somewhat higher values in the western and south-western part of the scene. "

10. *Final sentence of Section 4.2.2. "The differences from the direct comparison between SPEX and RSP are somewhat larger than those from individual comparisons with HSRL-2 of SPEX and RSP, respectively. This suggests that the differences with HSRL-2 are not caused by common assumptions in the SPEX and RSP retrievals, but are rather caused by errors that are specific to each MAP". I don't follow the logic.*

Response:
If the differences with HSRL-2 are caused by common assumptions in the SPEX and RSP retrievals, the differences should be smaller when comparing SPEX and RSP (than comparing MAPs with HSRL-2) because the common assumptions should have little effect when comparing SPEX and RSP. However, the comparison between SPEX and RSP is worse than comparing MAPs with HSRL-2. Thus, we reach the statement " This suggests that the differences with HSRL-2 are not caused by common assumptions in the SPEX and RSP retrievals, but are rather caused by errors that are specific to each MAP"

11. *Section 4.2.3. page 13. Lines 15-17. "It should be noted that the smoke plume exhibits large spatial variation so part of the MAP-lidar differences can be attributed to the fact that different instruments see a slightly different part of the smoke plume". What about different angles from the same instrument seeing different parts of the smoke, or what if the smoke changes between the fore and aft angles are measured? What happens to the retrieval? It would be really nice to have a quantitative sense of how variable that plume is. Could we see a spatial plot of the smoke retrievals or at least have stdev on the parameters shown in Table 2.*

    Response:
    If the different viewing angles would see different parts of the smoke plume, this would result in a large $\chi^2$ of differences between forward model and measurements. This is not the case for the points in the Figure 7.

    To illustrate the spatial variability of the smoke plume, we have included Figure 7a, which gives a sense of how variable the AOD of smoke plume is. We have also included the standard deviation inside the plume from SPEX and RSP on the parameters shown in the Table 3.

12. *Page 13. Lines 25-26. "Our explanation for this, is that at high AOD the measured radiance and DoLP are less affected by the co-registration errors between viewing angles than for low AOD". How could this be? The evolving, hetereogeneous smoke plume has to be more difficult to co-register between angles than the unmoving ground.*

    Response:
    The land surface can be very patchy, especially near Fresno and Bakersfield. This leads to higher spatial variability in the radiance than the spatial variation of the smoke plume at 100 meter scale. The difference in co-location between the MAPs and HSRL-2 sampling may however be 1 km.

13. *Page 13. Lines 33-34. On the other hand, I think this is a really good explanation: "A possible explanation for the difference could be the simplified description of non- spherical particles in our retrieval approach. "*

    Response:
    Thanks.

14. *Figure 7d-f. Are lidar ratios here retrieved via HSRL method, or assumed? If assumed,*

*does these figures make any sense. If retrieved, then why not use retrieved throughout the paper? Or show that they are worse than assumed. This whole retrieved vs. assumed lidar ratio choice never sat well with me throughout the manuscript.*

Response:

Shown in the plots are those retrieved from the HSRL method. This is consistent with our comparison for the smoke case for high AOD from the HSRL method. The only place we use the assumed lidar ratio is for the comparison of AOD in the low AOD case. For these cases we do not compare the lidar ratio because indeed that would not make sense.

15. *Table 2. Maybe show stdev along with mean? Or show spatial distribution if any of these properties are varying downwind?*

Response:

We have done so in the revised manuscript. Please see Figure 7a and Table 3.

16. *Page 14. Line 13. "the latter value is closer to the ALH derived from HSRL-2 (2.64 km)". Sure slightly closer, but still 1 km off. Not that much different from SPEX.*

Response:

We agree. We re-wrote the phrase to the paper in Sect. 4.2.4:

" For the Aerosol Layer Height (ALH), SPEX retrieves a higher value (4.417 km) than RSP (1.148 km), where the latter value is somewhat closer to the ALH derived from HSRL-2 (2.64 km). "

17. *Page 14. Line 14. The explanation of ALH being difficult to retrieve without UV might be elaborated on a little here.*

Response:

We added a phrase to the paper in Sect. 4.2.4:

" Here, it should be noted that for SPEX the shortest wavelength that is used in the retrieval is 450 nm, so we do not expect an accurate ALH retrieval because the retrieval of ALH from polarization requires a strong signal from Rayleigh scattering (Wu et al., 2016). "

18. *Finally. . . don't you want to state a conclusion? What is the overarching thing you have learned? If this were my paper I would conclude that the 3 polarimeters are producing*

*comparable results when forced through the same algorithm. The exception being aerosol layer height and perhaps some coarse mode parameters, which suffer from not having the bands that these parameters are sensitive to: shortwave (410 nm) and SWIR, respectively. So when there is no sensitivity, the retrieval becomes a random number generator. But for parameters that the instruments are sensitive to, there is little difference between instruments. It is still TBD whether algorithmic differences are going to matter. But it is not my paper. The authors can choose to write a conclusion of their choice. Or not.*

Response:

Thanks. We included these summaries to the paper in the conclusion part:

" In this study, 3 polarimeters produced comparable results when using the same algorithm. The exception were the ALH and some coarse mode parameters, which were mainly caused by not having the bands that these parameters were sensitive to: shortwave (410 nm) and SWIR, respectively. For parameters that the instruments were sensitive to, good agreements were found among instruments. Our results corroborate the findings of earlier studies that different combinations of spectral and angular measurements yield a very similar retrieval capability for aerosol properties (Hasekamp and Landgraf, 2007; Wu et al., 2015; Hasekamp et al., 2019) "

**References**

Dubovik, O., Holben, B. N., Lapyonok, T., Sinyuk, A., Mishchenko, M. I., Yang, P., and Slutsker, I.: Non-spherical aerosol retrieval method employing light scattering by spheroids, Geophys. Res. Lett., 29, 1415+, https://doi.org/10.1029/2001gl014506, 2002.

Eck, T. F., Holben, B. N., Reid, J. S., Dubovik, O., Smirnov, A., O'Neill, N. T., Slutsker, I., and Kinne, S.: Wavelength dependence of the optical depth of biomass burning, urban, and desert dust aerosols, Journal of Geophysical Research: Atmospheres, 104, 31 333–31 349, URL https://agupubs.onlinelibrary.wiley.com/doi/abs/10.1029/1999JD900923, 1999.

Hasekamp, O. P. and Landgraf, J.: Retrieval of aerosol properties over land surfaces: capabilities of multiple-viewing-angle intensity and polarization measurements, Appl. Opt., 46, 3332–3344, https://doi.org/10.1364/ao.46.003332, 2007.

Hasekamp, O. P., Fu, G., Rusli, S. P., Wu, L., Di Noia, A., Brugh, J. a. d., Landgraf, J., Martijn Smit, J., Rietjens, J., and van Amerongen, A.: Aerosol measurements by SPEX-one on the NASA PACE mission: expected retrieval capabilities, Journal of Quantitative Spectroscopy and Radiative Transfer, 227, 170–184, URL http://www.sciencedirect.com/science/article/pii/S0022407318308653, 2019.

Holben, B. N., Tanrĺ, D., Smirnov, A., Eck, T. F., Slutsker, I., Abuhassan, N., Newcomb, W. W., Schafer, J. S., Chatenet, B., Lavenu, F., Kaufman, Y. J., Castle, J. V., Setzer, A., Markham, B., Clark, D., Frouin, R., Halthore, R., Karneli, A., O'Neill, N. T., Pietras, C., Pinker, R. T., Voss, K., and Zibordi, G.: An emerging ground-based aerosol climatology: Aerosol optical depth from AERONET, J. Geophys. Res., 106, 12–12 097, https://doi.org/10.1029/2001jd900014, 2001.

O'Neill, N. T., Eck, T. F., Smirnov, A., Holben, B. N., and Thulasiraman, S.: Spectral discrimination of coarse and fine mode optical depth, Journal of Geophysical Research: Atmospheres, 108, URL https://agupubs.onlinelibrary.wiley.com/doi/abs/10.1029/2002JD002975, 2003.

Wu, L., Hasekamp, O., van Diedenhoven, B., and Cairns, B.: Aerosol retrieval from multiangle, multispectral photopolarimetric measurements: importance of spectral range and angular resolution, Atmospheric Measurement Techniques, 8, 2625–2638, https://doi.org/10.5194/amt-8-2625-2015, 2015.

Wu, L., Hasekamp, O., van Diedenhoven, B., Cairns, B., Yorks, J. E., and Chowdhary, J.: Passive remote sensing of aerosol layer height using near-UV multiangle polarization measurements, Geophys. Res. Lett., 43, 8783–8790, https://doi.org/10.1002/2016gl069848, 2016.

---

## Author Comment (AC3) · 6 Nov 2019

**Reply to General comments**:

1. *When the authors introduce the SRON multimode retrieval algorithm in section 2.1, no aerosol size distribution parameters are included in the state vector. However, in the retrieval results, effective radius of fine and coarse mode particles are shown. Although the calculation of fine and coarse mode effective radius is presented in section 2.2, the retrieved aerosol parameters related to size parameters are not clear.*

   Response:
   Thanks. We added a description of the multimode retrieval algorithm to the paper in Sect. 2.1:
   " In principle, the idea of the multimode approach is that instead of fitting the size distribution parameters (the effective radius $r_{\mathrm{eff}}$ and the effective variance $v_{\mathrm{eff}}$) of two modes, one aims to fit the size distribution with a larger number of modes for which $r_{\mathrm{eff}}$ and $v_{\mathrm{eff}}$ are fixed. The advantage of this approach is that it makes the inversion problem more linear since $r_{\mathrm{eff}}$ and $v_{\mathrm{eff}}$ tend to make the inversion highly nonlinear. Another advantage is that the multimode approach has more freedom in fitting different shapes of size distribution if the number of chosen modes is sufficiently large. In this paper, multimode retrievals based on 5 modes are used and the aerosol size distribution are described in Table 3 (Fu and Hasekamp, 2018). "

   In the retrieval, we don't retrieve $r_{\mathrm{eff}}$ and $v_{\mathrm{eff}}$ for the 5 modes. The fine and coarse effective radius are calculated after retrievals based on Sect. 2.2. The sensitivities of the retrieved aerosol parameters related to different particle size parameters (parametric 2-mode, 3 to 10 multimode) have been extensively studied in Fu and Hasekamp (2018).

2. *As defined in the manuscript, the $\chi^2$ used to decide retrieval convergence is different for different instruments. For example, for AirMSPI, observed intensities in 8 bands and DoLP in 3 bands are used in the retrieval, while radiance and DoLP at 16 wavelengths for SPEX are used. Although $\chi^2$ is defined as a mean value of total number of measurements, the ratios $(\frac{F_i - y_i}{S_y(i,i)})$ in Eq. 3 for radiance and DoLP may have different scales. Therefore, if different numbers of radiances and DoLP are used even though two instruments have the same total number of measurements, the $\chi^2$ may differ a lot. Does this problem affect the retrieval results between different instruments? Do they use the same threshold $\chi^2_{\max}$?*

   Response:
   In this paper, for different instruments we use the same threshold $\chi^2_{\max} = 1.5$. It is true that for the different instruments there are different contributions to the $\chi^2$. This would only pose a problem if the assumed errors in $S_y$ are a poor representation of the true measurement errors. We believe we have used reasonable error estimates in our $S_y$ for the

differnt instruments so this should not pose a problem.

3. *The retrieval results of 3 different instruments are compared in this manuscript, but only some statistical parameters, such as MAE, bias and STD are presented. Are there any conclusions or suggestions about the measurements (radiance or DoLP) at which wavelengths are combined better for aerosol retrieval? Or are different numbers of multi-angle measurements affect aerosol retrievals a lot? I think more similar common summaries could attract audiences.*

Response:
Our study confirms earlier studies that different combinations of spectral and angular measurements yield a very similar retrieval capability for aerosol properties (Hasekamp and Landgraf, 2007; Wu et al., 2015; Hasekamp et al., 2019). We have highlighted this in the conclusion of the revised manuscript.

4. *In the state vector, aerosol column numbers and microphysical properties are included, thus the AOD in the retrieval at different wavelengths are calculated from retrieved column numbers and other parameters. I'm a little confused that why the authors use different wavelengths when compare total AOD and fine and coarse modes AOD (Figure 1 and Figure 3). If the same wavelengths are used, the retrieval performance of fine, coarse mode AOD and total AOD can also be evaluated.*

Response:
Only measurements at 500 nm have been used to compare the fine and coarse mode AOD because the measurements (fine and coarse mode AOD) at other wavelengths are not available in the SDA product.

5. *The surface reflectance parameters are retrieved simultaneously with aerosol properties in the algorithm. How is the performance of surface reflectance retrieval in the campaign? Are the accuracies of retrieved aerosol properties related to surface reflectance?*

Response:
We do not have a good reference to evaluate the accuracy of the retrieved surface parameters. Instead, we have evaluated the difference between MAP and HSRL-2 as function of retrieved surface properties. The results are show in Figure R1 of this response. We do not see clear correlation with surface parameters here.

[Figure]

Figure R 1: **Sensitivities between AOD differences (between MAPs and HSRL-2) and surface parameters. (a)-(h)** the low AOD case. **(i)-(p)** the high AOD (smoke) case. 1st, 2nd, 3rd, and 4th column respectively represent results with respect to BRDF scaling parameters for wavelength bands ($A_{532}$), Parameter 1 of RPV model ($g$), Parameter 2 of RPV model ($k$), and Scaling parameter for polarized reflectance ($B$).

6. *The retrieval accuracy of fine and coarse mode AOD depend on the retrieved aerosol microphysical properties. If the dependence of the retrieval bias of $\tau^f$ and $\tau^c$ on the accuracy of retrieved $r_{\mathrm{eff}}$ or refractive index is shown, it will be interesting and beneficial for distinguishing aerosol types.*

Response:
We do not see such dependencies in the available data and also we do not really expect it.

**Reply to Specific comments**:

1. *In the introduction part, the third paragraph in page 2 indicates that combining both intensity and polarization measurements at multiple viewing angles is beneficial for aerosol retrieval. However, this paragraph is too short and simple. This is the most important feature of 3 MAPs used in this manuscript to do retrieval. I think more theoretical foundation and how previous studies use these information could be added.*

Response:
We extended the paragraph 3 (in the introduction) in the paper by adding a review of previous studies:

" The reason is that the angular dependence of the scattering matrix elements related to linear polarization, depend strongly on the microphysical aerosol properties, like refractive index and particle size (Hansen and Travis, 1974; Mishchenko and Travis, 1997). Furthermore, the polarization signal is mostly dominated by light that has been scattered only once, which means that the characteristics of the scattering matrix remain largely preserved in a top-of-atmosphere polarization measurement. The added value of polarization has been demonstrated by a number of studies on synthetic measurements (Mishchenko and Travis, 1997; Hasekamp and Landgraf, 2007; Hasekamp, 2010; Knobelspiesse et al., 2012), airborne measurements (Chowdhary et al., 2005; Waquet et al., 2009; Xu et al., 2017; Wu et al., 2015, 2016), and spaceborne measurements (Hasekamp et al., 2011; Dubovik et al., 2011; Fu and Hasekamp, 2018). These algorithms can be divided in two main groups: LookUp-Table (LUT) based approaches and full inversion approaches. Generally speaking, LUT approaches are faster but less accurate than full inversion approaches because LUT approaches choose the best fitting aerosol model from a discrete lookup table. Full inversion approaches are more accurate but slower because they require radiative transfer calculations as part of the retrieval procedure. The LUT algorithms are e.g., the LOA LUT algorithm over ocean (Deuzé et al., 2000), the LOA LUT algorithm over land (Deuzé et al., 2001; Herman et al., 1997), and the SSA LUT algorithm (Waquet et al., 2016). The full inversion algorithms are e.g., the GRASP algorithm (Dubovik et al., 2011), the SRON-Aerosol algorithm (Hasekamp and Landgraf, 2007; Hasekamp et al., 2011; Stap et al., 2015;

Wu et al., 2015, 2016; Di Noia et al., 2017; Fu and Hasekamp, 2018), the JPL algorithm (Xu et al., 2017), the GISS algorithm (Waquet et al., 2009) and the MAPP algorithm (Stamnes et al., 2018). Besides, some additional aerosol retrieval approaches can be found in (Sano et al., 2006; Cheng et al., 2011; Masuda et al., 2000; Lebsock et al., 2007). It should be noted that of the full inversion approaches only the SRON-Aerosol algorithm and the GRASP algorithm have been applied at a global scale. "

We also included more theoretical description for the retrieval algorithm in Sect. 2.1 from Eq. (4) to (6).

2. *The paragraph at page 3 line 6-10 has little relationship with this study. I believe the authors could delete or short this paragraph and combine it with last paragraph.*

Response:

Thanks. We have shorted this paragraph and combined it with the previous paragraph, as stated in the end of paragraph 4 of introduction:

" The POLDER design also forms the blueprint for the 3MI instruments (Fougnie et al., 2018), to be flown on METOP-SG in the time frame ∼2020-2035. "

3. *When giving the information of ACEPOL campaign in the introduction, the information about the altitude aircraft flying is suggested to be provided due to the retrieval of ALH, especially at smoke plume case whose ALH is always high.*

Response:

We agree. We added the altitude of the NASA ER-2 flight in the introduction: " All 4 airborne MAPs listed above were mounted on the NASA Earth Resources-2 (ER-2) high altitude (∼20 km) aircraft (Navarro, 2007) during the Aerosol Characterization from Polarimeter and Lidar (ACEPOL) campaign, which was performed from October-November 2017, starting from the NASA Armstrong airbase in Palmdale, California. "

4. *At page 4 line 20, the meaning of $k$ in the equation is not explained.*

Response:

Thanks. We added it to the paper in Sect. 2.1:

" where $k$ is a parameter that varies between 0 and 1. This parameter controls the slope of the reflectance with respect to the illumination and view angles (Rahman et al., 1993). "

5. *At page 11 line 18-19, the authors present "the MAE gets smaller with increasing wavelengths, which is mainly caused by the fact that AOD value itself decreases with wavelength". Some other parameters such as mean relative error (MRE) or root mean squire error (RMSE) could remove this effect and are recommended to be compared.*

   Response:
   Thanks. Yes, the MRE can remove this effect, but the RMSE not. We added the MRE to the paper for all the AOD comparisons with AERONET and HSRL-2, and indeed see that the MRE does not decrease with wavelength.

6. *The sentences at line 22-23 and line 30-31 in page 11 present the same thing.*

   Response:
   We re-wrote the latter one (which is especially for the coarse mode effective radius) to the paper in Sect. 4.1:
   " This is in line with synthetic studies (e.g., Hasekamp et al. (2019)) that $r_{\mathrm{eff}}^{\mathrm{c}}$ is a difficult parameter to retrieve, in particular for small AOD values. "

7. *At page 13 line 1-2, "for low AOD the effect of the surface on the measured radiances is larger than for SPEX airborne" is presented. I'm a little confused why.*

   Response:
   We re-wrote this sentence to:
   " A possible explanation is that for low AOD the radiance and polarization measurements have strong influence from the spatially inhomogeneous surface, and therefore errors due to inter-angle mis-registration, which are larger for RSP than for SPEX, may be significant. "

8. *At page 14 line 13-14, the authors explained that the shortest wavelength for SPEX is 450 nm and not suitable for ALH retrieval. Do you mean the shorter wavelengths such as UV band benefit ALH retrieval? More clear and straight forward sentences are suggested to be used. Moreover, this explanation for ALH retrieval is too simple and this may be only one of many reasons. I believe reading more related papers about ALH retrieval could help the authors explain this problem more clearly and deeply.*

   Response:
   To our best knowledge, Wu et al. (2016) is the only paper for ALH retrieval from MAP measurements. We extended the explanation in Sect. 4.2.4 by adding:

" Here, it should be noted that for SPEX the shortest wavelength that is used in the retrieval is 450 nm, so we do not expect an accurate ALH retrieval because the retrieval of ALH from polarization requires a strong signal from Rayleigh scattering (Wu et al., 2016). "

9. *Some sentences in this manuscript are a little complex and confused, especially in section 1 and section 4. More concise sentences are recommended.*

Response:
Thanks. We believe both Section 1 and 4 have been improved in the new version paper.

**References**

[revised manuscript text omitted]

---

## Author Comment (AC4) · 6 Nov 2019

**Reply to general comments**:

1. *Although the paper focuses on aerosol retrievals, surface is an important component in the retrieval process and is included in the state vector. A good characterization of surface reflectance can significantly affect the retrieval accuracy of aerosol properties, which is especially true when aerosol loading is small (such as of the most ACEPOL cases). So, as a reader I would like to see some retrieval results for surface BRDF/BPDF properties and how the retrievals behave between different polarimeters.*

    Response:
    Thanks. We included a figure (Figure 6) showing the dependence of AOD difference between MAP and HSRL2 as function of the surface reflection (A) at 532 nm in the revised manuscript.

2. *The retrieval algorithm needs some more clarification in a few aspects of the radiative transfer calculations and the inversion configurations. These include: (i) which radiative transfer model and what are the relevant assumptions (such gas absorptions, Rayleigh scatterings, etc) in the radiative transfer assumptions? (ii) How the first guess of the state vector is defined? While the first guesses for aerosol parameters are mostly given, the paper mentions nothing about prior values for surface BRDF/BPDF parameters. (iii) It is not clear how the aerosol refractive index are treated, although it is mentioned to use the D'Almeida et al (1991) database. (iv) It is also not clear about how the weighting matrix (W) in the cost function is defined, as well as the threshold for the goodness of fit. Please refer to the relevant specific comments below for more details.*

    Response:
    - (i) We use the SRON radiative transfer mode LINTRAN (Hasekamp and Landgraf, 2005; Schepers et al., 2014). Rayleigh scattering cross sections are from Bucholtz (1995). Values for O3, NO2, and H2O columns are taken from MERRA-2 and AFGL as mentioned on page 11.
    -(ii) The first guess is obtained using a LUT approach. We extended the description. The LUT retrieval provides first guess values for aerosol and surface properties. The LUT retrieval itself starts with fixed values for all surface properties, i.e., 0.05, -0.09, 0.80, 1.0 for $A$, $g$, $k$, $B$, respectively. The prior values are listed in Table 2 of the revised manuscript.
    -(iii) The treatment of the refractive index is explained end page 4 and start page 5. The coefficients are included in the state vector.
    -(iv) We include the values for the weighting matrix in Table 2 of the revise manuscript.

3. *By reading the title of the article (Aerosol retrievals from the ACEPOL campaign), I would expect to see aerosol retrievals from different polarimeters and from their respective*

*aerosol products. Are there any aerosol products available from the ACEPOL campaign with other existing retrieval algorithms? If yes, it would be more helpful to compare the aerosol retrievals from different algorithms. Such a comparison may also explain the consistent biases in the retrieved aerosol size (Figure 2), depolarization ratio and lidar ratio (Figure 7). Otherwise, I would suggest to make the article title more specific, for instance, by adding "using the SRON algorithm".*

Response:

We agree maybe the title is too general, and we have changed the title to:

" Aerosol retrievals from different polarimeters during the ACEPOL campaign using a common retrieval algorithm "

**Reply to specific comments**:

1. *Page 4, first paragraph of section 2.1. Description about aerosol refractive index is too brief. Please clarify: (i) at which relative humidity (RH) is assumed for the D'Almeida et (1991) database, or a dynamic RH relationship is considered with ancillary meteorological data? This is important as the inorganic aerosols are strongly hygroscopic. (ii) How the coefficients are defined for combining the aerosol species? In terms of volume concentrations? (iii) Are the different aerosol species internally or externally mixed in the calculation of modal refractive index? In addition, it would be helpful if the refractive indices used in this study being provided in a supplemental document.*

Response:

(i) We only use the refractive index spectra from d'Almeida for the spectral dependendence of the refractive index. So. no assumtions are needed on RH.

For the comment (ii) we have added more description to the paper in Sect. 2.1:

" The coefficients $\alpha_k$ are the real numbers between 0 and 1, and are defined as weighting factors to combine the refractive index spectra for different aerosol components, e.g., DUST, water (H2O), Black Carbon (BC), INORGanic matter (INORG). In this study, we set $n_\alpha = 2$ and assume that spectral dependence of the fine mode and the coarse mode refractive indices can be described respectively by INORG+BC and DUST+INORG. Note that this assumption is flexible and can be updated according to the information content of the measurement. Also spectra based on Principal Component Analysis (PCA) can be used as in Wu et al. (2015). The standard refractive index spectra are only used to describe the spectral dependence as the MAP measurements do not contain sufficient information to retrieve the refractive index for each wavelength separately. "

Comment (iii):
We compute the refractive index given the formula on top of page 5 of the revised manuscript. Given that we perform one Mie/T-matrix computation per mode for one refractive index, this implicitly assumes internal mixing.

2. *Page 5, line 6. Please give the explicit expression for $R(G)$.*

Response:
In the new version, we included $R(G) = \frac{1-A(\lambda)}{1+G}$ in the Sect. 2.1.

3. *Section 2.1. The number of elements in state vector for different sensors would be different because of the different number of spectral bands. I would recommend include a table to list the detailed elements (and numbers) of the retrieved parameters for individual polarimeters. Correspondingly, the selected bands and number of angles for each observation set (as described in section 3.1-3.3) can also be listed in the same table. This will give the reader a clearer picture about the retrieval configuration for different sensors.*

Response:
Thanks. We have included Table 2 as suggested, which lists the viewing angles and wavelengths used in retrievals among SPEX airborne, RSP, and AirMSPI. The state vectors corresponding to these three polarimeters are also listed in the table. For the state vector, the only difference among three instruments is the BRDF scaling parameter $A(\lambda)$ which is wavelength-dependent.

4. *Section 2.1. It is not mentioned in algorithm description about: (i) what radiative transfer model is used and how many layers of atmosphere is assumed; (ii) how the gas absorption are treated; (iii) How the Rayleigh scattering are calculated. Please clarify.*

Response:
We have clarified these aspects in Sect. 2.1: " **F** consists of a radiative transfer model, for which we use the SRON radiative transfer model LINTRAN Landgraf et al. (2001); Hasekamp and Landgraf (2002, 2005); Schepers et al. (2014). All the radiative transfer calculations are performed for a model atmosphere that includes Rayleigh scattering, scattering and absorption by aerosols, and gas absorption. Rayleigh sctterring cross sections are used from Bucholtz (1995). The forward model simulates Stokes parameters $I, Q, U$ at the top of the atmosphere (800 km) or the height of the research flight (e.g., ∼20 km for NASA ER-2 in this paper) for given optical properties (scattering and absorption optical thickness and scattering phase matrix for each vertical layer of the model atmosphere ( 15

layers of atmosphere is assumed  ). The other part of the forward model computes the optical properties from the aerosol microphysical properties using the tabulated kernels of Dubovik et al. (2006) for a mixture of spheroids and spheres. ”

5. *Page 5, Equation (2). Please clarify how the weight matrix (W) is defined to regulate the ranges of individual state parameters.*

   Response:
   Table 2 of the revised manuscript gives the elements of the weighting matrix. It has a comparable role as the prior covariance matrix in Optimal Estimation, except that for our inversion we have an additional regularization parameter that scales the whole matrix.

6. *Page 5, Equation (2). It is not clear how the prior state vector is defined for surface parameters. Please clarify.*

   Response:
   For the comment 5 and 6, we added a phrase to the paper in Sect. 2.1:
   “ Table 2 shows the values in $\vec{x}_a$ including the prior values for aerosol and surface parameters. $\mathbf{W}$ is a diagonal matrix and its diagonal values are also shown in Table 2 (in the “weight” column).  ”
   Table 2 is included in the new version paper.

7. *Page 5, line 15. It is mentioned here “Stokes parameters I, Q, U at the top of the atmosphere” are simulated, but it is not clear what is the TOA altitude as defined. Moreover, the ACEPOL measurements are taken at an altitude of the ER-2 flights. The radiative transfer model should simulate the radiances as observed at the flight level. Please justify.*

   Response:
   Yes. We have added information to avoid confusion in Sect. 2.1:
   “ The forward model simulates Stokes parameters $I, Q, U$ at the height of the observation (e.g., $\sim 20\,$km for NASA ER-2 in this paper)  ... ”

8. *Page 5, line 29. Is a constant threshold for Kai-Square used for all retrievals across different instruments? Please clarify.*

   Response:
   Yes, we use 1.5 as the threshold for all instruments and for all retrieval cases in the paper.

This was already mentioned in the Sect. 4.

9. *Page 6, Equation (4). The symbol "G" is already used in equation (1) to denote hot-spot geometry factor. A different symbol should be used to avoid ambiguity.*

Response:
Thanks. We have used "O" to replace "G" here.

10. *Page 6, Equation (7). Are there any references for calculating the columnar depolarization ratio in this way? I recall some studies (sorry I couldn't find the paper) used layer extinction coefficient (rather than backscatter coefficient) as the weighting parameter.*

Response:
Actually, either the extinction coefficient or the backscatter coefficient can be taken as the weighting parameter. The reason why we use backscatter coefficient here is because for ACEPOL, the backscatter profiles from HSRL-2 are more accurate than the extinction profiles from HSRL-2.

11. *Page 9, line 24. Do you meant to "Where the HSRL method is NOT available for the extinction products . . .."*

Response:
We have changed that part in the new version to avoid confusion:
" For ACEPOL, the extinction products from the HSRL method are reported at 150 m vertical resolution and at temporal resolution of 60 s generally and 10 s. Additionally, the aerosol extinction products at 355 nm and 532 nm are also provided based on the aerosol backscatter and an assumed lidar ratio of 40 sr, and reported at the backscatter resolution. "

12. *Page 11, line 32. It seems the effective radius for coarse modes 4 and 5 are much smaller than the AERONET climatology as reported in Dubovik et al (2002). So why not define a large effective radius values for these two modes.*
*Reference: Dubovik, O. et al (2002), Variability of Absorption and Optical Properties of Key Aerosol Types Observed in Worldwide Locations, Journal of the Atmospheric Sciences, 59(3), 590-608.*

Response:
Yes, this is also possible. Actually we have other options for multimode retrievals as shown

in Table 2 of Fu and Hasekamp (2018). For example, in the 7-mode retrieval, the largest effective radius is $3.0\,\mu m$. We can also re-define another 5 modes with larger effective radius for coarse modes 4 and 5. But for the 5-mode retrieval used in this paper, given that all the parameters seem to be well retrieved except for the coarse mode AOD (biased with AERONET SDA data) which is very small for the ACEPOL campaign, we think the current 5 modes are still reliable for this study.

13. *Figure 28. Authors may consider to replace the background of Figure 28a with a true color image of the smoke plume. I have seen such a figure from AirHARP gallery. It would be even better if a retrieved AOD map for the smoke plume is presented here.*

Response:
For the true color image of the smoke plume, we don't have it. We included Figure 7a on SPEX spatial sampling, which gives a sense of how variable the smoke plume is. Figure 7a is the retrieved AOD map for the smoke plumn.

14. *Page 13, line 23-24. It is mentioned here the smoke plume has large spatial variability that may contribute to the retrieval uncertainty. The suggestion above (#13) would at least give a visual expression how large the spatial variability is. In addition, the MAP algorithm would have challenge to retrieve AOD as different view angles see different location (thus AOD) of the elevated plume due to the parallax displacement. Can the authors provide some insights on how to addressing this challenge in the retrieval?*

Response:
The different viewing angles see a slightly different location but the difference is on the order of 100 meter. We do not expect the AOD to vary drastically over this distance. The difference in sampling between the MAPs and HSRL-2 however, may be on the order of 1 km, which may affect the comparion, as AOD will show some variation over a distance of 1 km in the smoke plume. So, the variability is not so large that it affects the retrieval uncertainty but rather limits the comparison to HSRL2.

15. *Finally, I would like to see a figure of retrieved particle size distribution for the smoke case, which would help interpreting the retrieval results listed in Table 2.*

Response:
We agree, and have included Figure 9 in the paper for number particle size distribution

from SPEX and RSP in the smoke plume.

**References**

Bucholtz, A.: Rayleigh-scattering calculations for the terrestrial atmosphere, Applied Optics, 34, 2765–2773, URL https://www.osapublishing.org/ao/abstract.cfm?uri=ao-34-15-2765, 1995.

Dubovik, O., Sinyuk, A., Lapyonok, T., Holben, B. N., Mishchenko, M., Yang, P., Eck, T. F., Volten, H., Muñoz, O., Veihelmann, B., van der Zande, W. J., Leon, J. F., Sorokin, M., and Slutsker, I.: Application of spheroid models to account for aerosol particle nonsphericity in remote sensing of desert dust, Journal of Geophysical Research (Atmospheres), 111, D11 208+, https://doi.org/10.1029/2005jd006619, 2006.

Fu, G. and Hasekamp, O.: Retrieval of aerosol microphysical and optical properties over land using a multimode approach, Atmospheric Measurement Techniques, 11, 6627–6650, https://doi.org/10.5194/amt-11-6627-2018, 2018.

Hasekamp, O. P. and Landgraf, J.: A linearized vector radiative transfer model for atmospheric trace gas retrieval, J. Quant. Spec. Radiat. Transf., 75, 221–238, https://doi.org/10.1016/s0022-4073(01)00247-3, 2002.

Hasekamp, O. P. and Landgraf, J.: Retrieval of aerosol properties over the ocean from multi-spectral single-viewing-angle measurements of intensity and polarization: Retrieval approach, information content, and sensitivity study, J. Geophys. Res., 110, D20 207+, https://doi.org/10.1029/2005jd006212, 2005.

Landgraf, J., Hasekamp, O. P., Box, M. A., and Trautmann, T.: A linearized radiative transfer model for ozone profile retrieval using the analytical forward-adjoint perturbation theory approach, J. Geophys. Res., 106, 27+, https://doi.org/10.1029/2001jd000636, 2001.

Schepers, D., aan de Brugh, J. M. J., Hahne, P., Butz, A., Hasekamp, O. P., and Landgraf, J.: LINTRAN v2.0: A linearised vector radiative transfer model for efficient simulation of satellite-born nadir-viewing reflection measurements of cloudy atmospheres, Journal of Quantitative Spectroscopy and Radiative Transfer, 149, 347–359, URL http://www.sciencedirect.com/science/article/pii/S002240731400363X, 2014.

Wu, L., Hasekamp, O., van Diedenhoven, B., and Cairns, B.: Aerosol retrieval from multiangle, multispectral photopolarimetric measurements: importance of spectral range and angular resolution, Atmospheric Measurement Techniques, 8, 2625–2638, https://doi.org/10.5194/amt-8-2625-2015, 2015.

---

## Referee Report (RR1)

The authors give more information and improve the descriptions in the revised manuscript, provide good explanations for the comments from the reviewers as well, except a few points could be demonstrated more clearly as follows.

**Minor comments**

1. In the introduction, the last part of the third paragraph (Line 4-10, Page 3) is suggested to be more concise and clear. There is no need to enumerate all the retrieval algorithms together.

2. Considering the wavelengths and viewing angles of MAPs have been shown in Table 2, the authors do not need to mention again in the text of Section 3.

3. From the aerosol retrieval results of two case studies at Oct. 26 and Nov. 9 with different AOD, did the authors find any distinctions about aerosol type between two cases? MAP measurements may be helpful for distinguishing aerosol types.

---

## Author Response (AR2)

Dear Editor,

Herewith we submit the revised manuscript. First we would like to thank you and reviewers, and appreciate all the comments and suggestions. In the following we will give our responses to the comments.

Kind regards,
Guangliang Fu and Otto Hasekamp,
on behalf of all co-authors

(The revised manuscript is in the latter part of this pdf.)

**Reply to comments**:

1. *In the introduction, the last part of the third paragraph (Line 4-10, Page 3) is suggested to be more concise and clear. There is no need to enumerate all the retrieval algorithms together.*

   Response:
   Thanks. We re-wrote the phrase at the top of page 3.

2. *Considering the wavelengths and viewing angles of MAPs have been shown in Table 2, the authors do not need to mention again in the text of Section 3.*

   Response:
   We agree and have re-wrote the phrases (Marked as "red") in Section 3 by referring the wavelengths and viewing angles to Table 2.

3. *From the aerosol retrieval results of two case studies at Oct. 26 and Nov. 9 with different AOD, did the authors find any distinctions about aerosol type between two cases? MAP measurements may be helpful for distinguishing aerosol types.*

   Response:
   Thanks. We agree that MAP measurements are helpful for distinguishing aerosol types. For the smoke case (Nov 9), this aspect was discussed in Section 4.2.4. The reason why we didn't discuss on aerosol types in the non-smoke case (Oct 26) is that the AODs here

are generally very small, i.e., $\sim$90% of the AODs at 532 nm are smaller than 0.1 (see Figure 5b), and for such low AOD the information on microphysical properties is very limited..

The revised manuscript starts from next page.

[revised manuscript text omitted]

Hair, J. W., Hostetler, C. A., Cook, A. L., Harper, D. B., Ferrare, R. A., Mack, T. L., Welch, W., Izquierdo, L. R., and Hovis, F. E.: Airborne High Spectral Resolution Lidar for profiling aerosol optical properties, Appl. Opt., 47, 6734–6752, http://ao.osa.org/abstract.cfm?URI=ao-47-36-6734, 2008.

Hansen, J. E. and Travis, L. D.: Light scattering in planetary atmospheres, Space Science Reviews, 16, 527–610, https://doi.org/10.1007/BF00168069, https://doi.org/10.1007/BF00168069, 1974.

Hasekamp, O. P.: Capability of multi-viewing-angle photo-polarimetric measurements for the simultaneous retrieval of aerosol and cloud properties, Atmospheric Measurement Techniques, 3, 839–851, https://doi.org/10.5194/amt-3-839-2010, 2010.

Hasekamp, O. P. and Landgraf, J.: A linearized vector radiative transfer model for atmospheric trace gas retrieval, J. Quant. Spec. Radiat. Transf., 75, 221–238, https://doi.org/10.1016/s0022-4073(01)00247-3, 2002.

Hasekamp, O. P. and Landgraf, J.: Retrieval of aerosol properties over the ocean from multispectral single-viewing-angle measurements of intensity and polarization: Retrieval approach, information content, and sensitivity study, J. Geophys. Res., 110, D20 207+, https://doi.org/10.1029/2005jd006212, 2005.

Hasekamp, O. P. and Landgraf, J.: Retrieval of aerosol properties over land surfaces: capabilities of multiple-viewing-angle intensity and polarization measurements, Appl. Opt., 46, 3332–3344, https://doi.org/10.1364/ao.46.003332, 2007.

Hasekamp, O. P., Litvinov, P., and Butz, A.: Aerosol properties over the ocean from PARASOL multiangle photopolarimetric measurements, J. Geophys. Res., 116, D14 204+, https://doi.org/10.1029/2010jd015469, 2011a.

Hasekamp, O. P., Litvinov, P., and Butz, A.: Aerosol properties over the ocean from PARASOL multiangle photopolarimetric measurements, J. Geophys. Res., 116, D14 204+, https://doi.org/10.1029/2010jd015469, 2011b.

Hasekamp, O. P., Fu, G., Rusli, S. P., Wu, L., Di Noia, A., Brugh, J. a. d., Landgraf, J., Martijn Smit, J., Rietjens, J., and van Amerongen, A.: Aerosol measurements by SPEXone on the NASA PACE mission: expected retrieval capabilities, Journal of Quantitative Spectroscopy and Radiative Transfer, 227, 170–184, http://www.sciencedirect.com/science/article/pii/S0022407318308653, 2019a.

Hasekamp, O. P., Gryspeerdt, E., and Quaas, J.: Analysis of polarimetric satellite measurements suggests stronger cooling due to aerosol-cloud interactions, Nature Communications, 10, 1–7, https://www.nature.com/articles/s41467-019-13372-2, 2019b.

[revised manuscript text omitted]